# Bridging the Gap Between Promise and Performance for Microscaling FP4 Quantization

**Vage Egiazarian**\* **& Andrei Panferov**\*
Institute of Science and Technology Austria

**Denis Kuznedelev**\*
Yandex Research

**Shubhra Pandit, Alexandre Marques & Mark Kurtz**
Red Hat AI

**Saleh Ashkboos & Torsten Hoefler**
ETH Zürich

**Roberto L. Castro**\***, Eldar Kurtić & Dan Alistarh** [†]
Institute of Science and Technology Austria & Red Hat AI

## Abstract

The recent hardware-accelerated microscaling 4-bit floating-point formats such as MXFP4 and NVFP4, supported on NVIDIA and AMD GPUs, promise to revolutionize large language model (LLM) inference. Yet, their practical benefits remain unproven. We present the first comprehensive study of MXFP4 and NVFP4 for post-training quantization, revealing gaps between their promise and real-world performance. Our analysis shows that state-of-the-art methods struggle with FP4, due to two key issues: (1) NVFP4's small group size *provably* neutralizes traditional outlier mitigation techniques; (2) MXFP4's power-of-two scale quantization severely degrades accuracy due to high induced error. To bridge this gap, we introduce Micro-Rotated-GPTQ (MR-GPTQ), a variant of the classic GPTQ quantization algorithm that tailors the quantization process to FP4's unique properties, by using block-wise Hadamard transforms and format-specific optimizations. We support our proposal with a set of high-performance GPU kernels that enable the MR-GPTQ format with negligible overhead, by rotation fusion into the weights, and fast online computation of the activations. This leads to speedups vs. FP16 of up to 3.6x layer-wise, and 2.2x end-to-end on NVIDIA B200, and of 6x layer-wise and 4x end-to-end on RTX5090. Our extensive empirical evaluation demonstrates that MR-GPTQ matches or outperforms state-of-the-art accuracy, significantly boosting MXFP4, to the point where it nears that of NVFP4. We conclude that, while FP4 is not an automatic upgrade over INT4, format-specialized methods like MR-GPTQ can unlock a new frontier of accuracy-performance trade-offs.

## 1 Introduction

Post-training quantization (PTQ) [41; 21; 36] is one of the most well-researched areas in model compression, in which the objective is to take an existing pre-trained model and reduce its size or computation while preserving most of its accuracy. With the advent of large language models (LLMs), PTQ has become a highly-active research area, e.g., [21; 59; 3; 15; 50; 19; 51] with significant industry adoption and practical impact [29].

---

\*Equal contributions.
[†]Correspondence to: `dan.alistarh@ist.ac.at`.

In this paper, we focus on quantization using the recently-introduced microscaling floating-point precision formats, specifically MXFP4 [47] and NVFP4 [43]. In a nutshell, these formats work by grouping elements into blocks of 32 or 16 elements, respectively, quantized together with a shared scale; to reduce the storage overhead, the scales themselves are also compressed, to distinct 8-bit format: a standard sharing between Exponent and Mantissa bits (E4M3) for NVFP4, and E8M0—essentially, rounding scales to powers-of-two—for MXFP4. As such, the NVFP4 format trades off additional space (4.5 bits per element on average, relative to 4.25 bits for MXFP4), in favor of additional precision. The promise of these formats is two-fold: first, they are *claimed to be more accurate* than the prior-generation integer precision formats such as INT4 [40]. Second, they are *supported in hardware*: NVIDIA Blackwell GPUs support matrix multiplications across both NVFP and MXFP formats, whereas AMD GPUs will support MXFP4 [1]. Despite these developments, little is known about the accuracy of these formats on real models or their practical performance.

**Contributions.** In this paper, we provide a first thorough study of the accuracy and performance limitations of the NVFP4 and MXFP4 formats through the prism of current state-of-the-art quantization methods, coupled with computational support. We focus primarily on weight-and-activation quantization to 4-bits per parameter, and investigate the interaction between these new formats, real parameter distributions, and state-of-the-art quantization algorithms. Our main findings are:

- We begin with an analysis of quantization error induced by the NVFP4 and MXFP4 formats over both Laplace-like heavy-tailed distributions, which arise in real-world weights and activations [2; 17], and over Normal parameter distributions, arising when processing weights and activations via rotations in popular methods such as QuIP/QuIP# [6; 50] or QuaRot [4]. Interestingly, we can *prove analytically* and *show empirically* that rotations *improve* MXFP4 accuracy, but *hurt* NVFP4 accuracy when coupled with standard Round-to-Nearest (RTN) quantization.

- Based on this analysis, we propose a new variant of the GPTQ weight quantization algorithm [21], called Micro-Rotated-GPTQ (MR-GPTQ), explicitly designed to maximize accuracy across both MXFP4 and NVFP4. The algorithm employs Hadamard rotations at the group level to "normalize" weights and activations, but in a novel *block-wise fused* form, which, as we show, can be supported without any runtime overheads on Blackwell GPUs. In addition, MR-GPTQ introduces a new efficient variant of the activation re-ordering heuristic for GPTQ, along with format-specific scale search optimizations.

- We perform the first extensive study of NVFP4 and MXFP4 *practical accuracy*, across standard Llama-3 [18] and Qwen-3 [55] models of different sizes, evaluated on standard zero-shot tasks [23]. We investigate a broad set of compression methods, including RTN, GPTQ [21], SmoothQuant [54], QuaRot [4], and SpinQuant [38], as well as our new MR-GPTQ approach. Results show that: (1) both NVFP4 and MXFP4 are lossy, with MXFP4 inducing major accuracy drops ($\sim$ 10% relative), and (2) that existing techniques are not well-suited for these new formats, as they do not always outperform RTN. On the positive side, we show that GPTQ and the MR-GPTQ variant yield consistently good recovery for NVFP4. Moreover, MR-GPTQ works particularly well in conjunction with MXFP4, recovering accuracy within 1-2% of NVFP4. For large models, we show that both formats can recover up to 98-99% of the baseline FP16 accuracy.

- Our main technical contribution is a new set of GPU kernels specific to the Blackwell architecture called QuTLASS, showing that the "micro-rotation" component of MR-GPTQ can be supported without loss of performance relative to standard multiplications. Specifically, this comes in the form of a lightweight fused kernel for online rotation of the activations. Remarkably, our kernel for MXFP4 *can obtain higher throughput* than an *ideal* NVFP4 matrix multiplication. Our kernels obtain near-ideal layer-wise speedups for both B200 and RTX5090 GPUs, of 3.6x and 6x, respectively, leading to end-to-end inference speedups of 2x and 4x, respectively.

## 2 BACKGROUND ON MICROSCALING FLOATING-POINT FORMATS

**General Definition.** The microscaling MXFP4 and NVFP4 formats employ hierarchical quantization, where elements within a block share a common scale factor, enabling efficient hardware implementation. Given a tensor divided into one-dimensional groups, we define a **Microscaling Block Floating-Point (MFP)** representation as follows. The **Group Size (G)** is the number of elements in each group before quantization. The **Element Representation (E)** is the format used to represent the individual elements within each block. The **Scale Representation (S):** The format used to represent the scale values for each group.

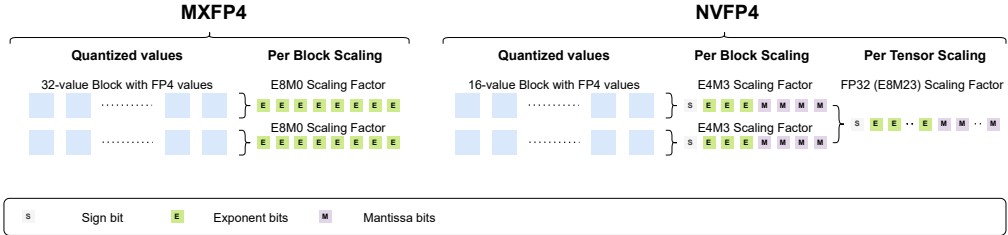

Figure 1: Schematic illustration of the MXFP4 (**left**) and NVFP4 (**right**) microscaling formats.

For floating-point (FP) formats, we use the notation **ExMy** to say that x bits are allocated to the exponent, and y bits are allocated to the mantissa. For instance, in the standard FP4 E2M1 representation, each FP4 element consists of 1 sign bit, 2 exponent bits, and 1 mantissa bit, providing 7 distinct positive values $\{0.5, 1.0, 1.5, 2.0, 3.0, 4.0, 6.0\}$ plus zero and the negatives.

**The MXFP4 (Microscaling FP4) Format.** This format [47] follows the specification ($G = 32, E =$ FP4, $S =$ E8M0). Its distinguishing features are the group size of 32 and its quantization of group scales to powers-of-two, given the use of E8M0, which dedicates all bits to the exponent and none to the mantissa. This design choice simplifies hardware multiplication; yet, as our experiments reveal, it often introduces quantization artifacts that can significantly impact model accuracy.

**The NVFP4 (NVIDIA FP4) Format** was introduced by NVIDIA for the Blackwell architecture [43], and employs a more flexible approach with ($G = 16, E =$ FP4, $S =$ E4M3). While sharing the FP4 element format with MXFP4, NVFP4 differs in two key aspects. First, it uses a 16-element group size, and, second, it uses a full FP8 representation for scales in E4M3, preserving more precise scaling information relative to E8M0. NVFP4 trades off a more accurate representation for weight and activation distributions, at the cost of increased bits-per-element (4.5 NVFP4 vs 4.25 for MXFP4).

**Related Work.** Early work on LLM quantization focused primarily on integer formats, with INT8 being the first to be investigated [13; 56], in conjunction with round-to-nearest (RTN) assignment over groups of consecutive weights and activations. FP formats introduce new possibilities but also challenges: while FP8 quantization is known to be near-lossless [29], the distribution of representable values in NVFP4/MXFP4 changes quantization dynamics. The GPTQ method [21] reached near-lossless INT4 compression via second-order weight adjustments. Its effectiveness for FP4 formats remains unexplored. Methods like AWQ [35], SqueezeLLM [28], and SpQR [15] relied on outlier-aware quantization strategies that assume uniform grids and large group sizes. The FP4 formats' small group sizes (16 or 32) and non-uniform grid inherently perform outlier mitigation, as we discuss in our analysis. Recent extreme compression techniques like QuIP [6], QuIP# [50] and QTIP [51] use rotation matrices to normalize the weight distributions. As we will see, this is not necessarily helpful for FP4 microscaling formats.

LLM activations are known to be extremely challenging to quantize, due to outlier features, defined roughly as elements up to 100× larger than average [13]. SmoothQuant [54] addresses this for INT8 by rescaling to redistribute outliers between weights and activations. Recent rotation-based methods like QuaRot [4] and SpinQuant [38] mitigate outliers through Hadamard transforms. In this paper, we discover novel trade-offs for these approaches.

Prior work investigating accuracy trade-offs under quantization, e.g., Yao et al. [56]; Liu et al. [37]; Huang et al. [27]; Gong et al. [25]; Li et al. [34]; Gong et al. [24]; Lee et al. [32]; Kurtic et al. [29] focuses almost exclusively on INT quantization. Despite industry claims about FP4's accuracy superiority [40; 43], rigorous evaluation remains absent so far, likely due to the recent introduction of this format. Our work addresses this gap.

## 3 A QUANTIZATION ERROR ANALYSIS OF NVFP4 AND MXFP4

Prior work on quantization [41; 13; 16] identified the average and top-element (outlier) mean-square error (MSE) as key quantities that can predict quantized model accuracy. In this section, we perform a model-based analysis of the NVFP4 and MXFP4 formats from the prism of these metrics.

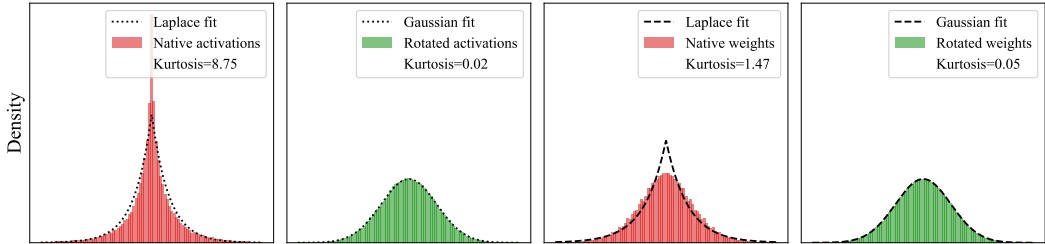

Figure 2: Distribution fits for aggregate weights and activations of Llama-3.1-8B-Instruct, with and without rotations. The Normal distribution is clearly a good fit for rotated weights and activations, while the Laplace distribution provides a good fit for the native distributions. Although native weights appear Normal, they have much heavier tails, as evidenced by the Kurtosis value.

**Modeling Distributions.** Early work on modeling LLM parameters assumed a Normal (Gaussian) distribution [14], consistent with common initialization schemes. Yet, more recent studies have identified that distributions with high kurtosis, such as the Laplace or Student-t distributions, better model the sharp peaks and outlier-prone tails of weights and activations [2; 17].

Here, we follow the latter line of work and model weights and activations as following a Laplace distribution. At the same time, interestingly, it can be proven that, *after the Hadamard rotation*, these tensors tend to follow a *normal* distribution [6; 50]. We empirically validate these findings via fits over common models, illustrated in Figure 2. Formally, our modeling is as follows:

**Definition 1** (Modeling). *We assume that the "native" weights and activations follow the Laplace distribution $W \sim \text{Laplace}(0, b)$ with density $f_W(w) = \frac{1}{2b} e^{-|w|/b}$, and variance $\text{Var}(W) = 2b^2$. We fix unit variance throughout, so $b = 1/\sqrt{2}$. The magnitude $Z = |W|$ is $\text{Exp}(\lambda)$ with rate $\lambda = 1/b = \sqrt{2}$, that is $f_Z(z) = \lambda e^{-\lambda z}$ and $F_Z(z) = 1 - e^{-\lambda z}$ for $z \geq 0$.*

*We assume that weights and activations rotated via the Hadamard transform follow a Normal distribution $V \sim \mathcal{N}(0, 1)$. The magnitude $Z = |V|$ is half-normal with $f_Z(z) = \sqrt{\frac{2}{\pi}} e^{-z^2/2}$ and $F_Z(z) = \text{erf}(z/\sqrt{2})$, $z \geq 0$, where where $\text{erf}(z)$ is the standard Gauss error function $\frac{2}{\sqrt{\pi}} \int_0^z e^{-t^2} dt$.*

**Quantization.** We model Microscaling Block Floating-Point (MFP) quantization as follows. Consider i.i.d. blocks containing $G \geq 2$ elements drawn from some distribution: $X = (X_1, \ldots, X_G)$ with $\text{Var}(X_i) = 1$ and $Z_i = |X_i|$. We assume a grid $\mathcal{Q} \subset [0, 1]$ that is finite, symmetric around 0, and includes both 0 and 1; we write $\mathcal{Q}^+ = \mathcal{Q} \cap [0, 1]$ and $q_{\min} = \min(\mathcal{Q}^+ \setminus \{0\})$. We use round-to-nearest (RTN) quantization, assuming probability 0 for rounding ties. Next, we formally define the scaling process. For simplicity, we will not *not* quantize the scale $s$ itself, and assume that values are normalized to $[-1, 1]$. We remove these assumptions in our numerical validation (Section 3.2).

**Definition 2** (Scales). *For a block of elements $X$, we define the unquantized scale $s := \max_{1 \leq i \leq G} |X_i|$, the normalized entries $U_i := X_i/s \in [-1, 1]$, the quantized normalized entries $\widehat{U}_i := \text{RTN}_\mathcal{Q}(U_i)$, and the de-normalized quantized values $\widehat{X}_i := s\,\widehat{U}_i$.*

**Definition 3** (Quantization Metrics). *For a group size $G$, we define: (i) The per-element MSE: $\text{MSE}(G) := \mathbb{E}[(X_1 - \widehat{X}_1)^2]$ (by symmetry, the choice of index can be arbitrary). (ii) The top-element MSE per block: Let $I_\star = \arg\max_{1 \leq i \leq G} |X_i|$, ignoring ties. Define $\text{MSE}_{\text{top}}(G) := \mathbb{E}\left[(X_{I_\star} - \widehat{X}_{I_\star})^2\right]$. We always use the same MFP map, i.e. same scale $s$, for both metrics.*

**Remark 1** (Quantization Dead-zone). *The first positive quantization level in the grid $Q$, which we denote by $q_{\min}$, induces the dead-zone half-width $\delta := q_{\min}/2$ on $[0, 1]$. If $|U_i| < \delta$, then $\widehat{U}_i = 0$.*

## 3.1 ANALYTICAL MSE BOUNDS

Next, we derive bounds on quantization error across top and average elements. First, notice that, in a simplified setting, applying the Hadamard rotation spreads the MSE evenly among elements.

**Lemma 1** (Top-Element MSE). *Assume a vector $x \in \mathbb{R}^G$ with coordinates i.i.d. $\mathcal{N}(0, 1)$, to which we apply a Hadamard rotation, perform MFP quantization in the $y$-domain to produce $\widehat{y}$, and reconstruct*

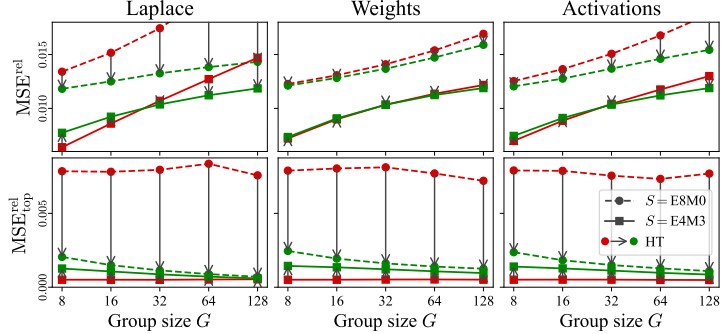 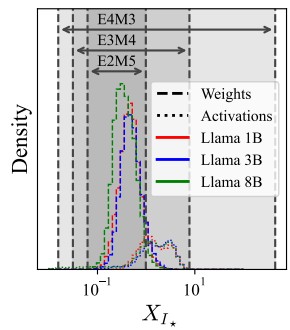

Figure 3: The effect of Hadamard Transform (HT) on MXFP4 (E8M0) and NVFP4 (E4M3) quantization on Laplace distribution samples and Llama-3.1-8B-Instruct weights and activations for various group sizes.

Figure 4: Ranges of FP8 scale format and observed weight and activation magnitudes.

$\widehat{x} = \frac{1}{\sqrt{G}} H^\top \widehat{y}$. *Define the quantization error vectors* $\varepsilon_y = \widehat{y} - y$ *and* $\varepsilon_x = \widehat{x} - x = \frac{1}{\sqrt{G}} H^\top \varepsilon_y$. *The expected squared error on the original top coordinate* $I_\star = \arg\max_i |x_i|$ *is the per-element MSE:*

$$\mathrm{MSE}_{\mathrm{top}}(G) = \mathbb{E}[(\varepsilon_x)^2_{I_\star}] = \frac{1}{G}\mathbb{E}\|\varepsilon_y\|_2^2 = \mathrm{MSE}(G).$$

**Remark 2** (Outlier preservation). *By contrast, it is immediate that* $\mathrm{MSE}_{\mathrm{top}}(G) = 0$ *in the absence of the Hadamard rotation, since we are doing* `absmax` *scaling, which preserves the top element.*

**Asymptotic MSE Analysis.** Thus, MSE is the key quantity we want to analyze. First, notice that, for any fixed grid with dead zone $\delta > 0$, for both Laplace and Normal models, $\lim_{G\to\infty} \mathrm{MSE}(G) = \mathrm{Var}(X_1) = 1$. Intuitively, this is because, as $G$ grows, the block maximum $M$ diverges, so $|U_1| = |X_1|/M \to 0$ in probability; the mass that survives the dead-zone vanishes. Consequently, the dominant part of the MSE $\mathbb{E}[(X_1 - \widehat{X}_1)^2]$ becomes $\mathbb{E}[X_1^2 \mathbf{1}\{|U_1| < \delta\}] \to \mathbb{E}[X_1^2] = 1$.

To get a more granular variant, we assume the large $G$ domain and examine the "preserved mass":

$$\mathcal{R}(G) := 1 - \mathrm{MSE}(G) = \mathbb{E}[X_1^2 \mathbf{1}\{|U_1| \geq \delta\}],$$

which captures the mass that *escapes* underflow. A precise calculation yields the following:

**Lemma 2** (Rates). *Let* $\delta = q_{\min}/2 \in (0, \frac{1}{2})$ *be the dead-zone halfwidth in the normalized domain.*

*For Laplace, we have:* $\mathcal{R}_{\mathrm{L}}(G) = \Theta\big((\log G)^2 G^{-\delta}\big)$, *and for Normal:* $\mathcal{R}_{\mathrm{N}}(G) = \Theta\big(\sqrt{\log G}\, G^{-\delta^2}\big)$.

**Discussion.** Since $0 < \delta^2 < \delta < 1$, we have that, for small $G$, the Laplace MSE should be below the MSE for the Normal distribution. Yet, for sufficiently large $G$, the Normal rate dominates the Laplace rate, meaning that $\mathrm{MSE}_{\mathrm{N}}(G) < \mathrm{MSE}_{\mathrm{L}}(G)$. As such, we predict a *crossover* phenomenon, where the MSE gap in favor of the (native) Laplace distribution will be *inverted* for larger group size $G$ in favor of the transformed Normal distribution. In short, transforms should *hurt* the original weights at small group sizes, and become effective as we increase it.

## 3.2 NUMERICAL VALIDATION

**Relative Errors.** In practice, the weight and activation distributions are *not of unit variance*. Shared scales give us control over the variance during the quantization process, but the aggregation of the proposed quadratic errors will be dominated by groups with higher variance. To address this, when analyzing real weights and activations, we use the relative version of the errors proposed above.

**Definition 4** (Relative Metrics). *Let* $I_\star = \arg\max_{1 \leq i \leq G} |X_i|$ *be the top group element. We define the* relative per-element MSE *as* $\mathrm{MSE}^{\mathrm{rel}}(G) := \mathbb{E}[\sum_{i=1}^{G}(X_i - \widehat{X}_i)^2 / \sum_{i=1}^{G} X_i^2]$, *and the* top-element MSE per block: $\mathrm{MSE}^{\mathrm{rel}}_{\mathrm{top}}(G) := \mathbb{E}[(X_{I_\star} - \widehat{X}_{I_\star})^2 / X_{I_\star}^2]$.

$\mathrm{MSE}^{\mathrm{rel}}$ is a key metric in compression theory [49]; in the context of LLM compression, Malinovskii et al. [39] to present a linear dependence between $\mathrm{MSE}^{\mathrm{rel}}$ and end-to-end accuracy decline. Additionally, recent lattice-based PTQ methods explicitly optimize for $\mathrm{MSE}^{\mathrm{rel}}$ when designing their

lattice [50; 51; 39]. For $\mathrm{MSE}_{\mathrm{top}}^{\mathrm{rel}}$, Lemma 3 shows how it accurately reflects the outliers' relative error as long as outliers are large, rare, randomly positioned, and $\mathrm{MSE}_{\mathrm{top}}^{\mathrm{rel}}$ is consistent for outliers and non-outliers (as shown by the shared scale quantization analysis below).

Figure 3 validates the analysis from Section 3.1 on samples from Laplace distribution, as well as on real weight and activation matrices from the Llama-3.1-8B-Instruct model. For $\mathrm{MSE}^{\mathrm{rel}}$ (top row) and NVFP4 ($G = 16$), the Hadamard Transform has a *negative effect* for small $G$ and a *positive effect* for larger $G$, exactly as predicted. To interpret the other effects, we have to better understand the effect of the shared scales quantization.

**Shared Scales Quantization.** Under fixed bit-width, microscaling floating point formats with a shared scale (stored, e.g., in E8M0 or E4M3) trade range for accuracy. We begin our analysis by examining the range required to fully cover weights and activations.

Figure 4 shows the logarithmic dynamic ranges of several FP8 formats and compares them with the empirical distributions of shared scales for weights and activations across multiple models. One can see that the dynamic range of $S = $ E4M3 covers the full range of these distributions. Trivially, $S = $ E8M0, having more range, can easily cover it too. When shared scales range is less than the dynamic range of $S$, they can always be represented by normal floating-point values with their relative error **(a)** bounded by $2^{-M}$ for mantissa precision $M$ and **(b)** translation-invariant to power-of-two shifts. For `absmax` quantization without rotations, this leads to $\mathrm{MSE}_{\mathrm{top}}^{\mathrm{rel}}$'s being insensitive to the shared scale magnitude in expectation over high dynamic range intervals, and, as the results, to $G$. We formalize this in Lemma 4.

This allows us to explain the effects of shared scale quantization on $\mathrm{MSE}_{\mathrm{top}}^{\mathrm{rel}}$ by relating it to the precision of the shared scales data type $S$ and the base data type $E$. We observe the following:

**(1) For MXFP4, top values inherit their precision from the base data type, and not the shared scale data type**. This is because $S = $ E8M0 is *coarser* than $E = $ E2M1, leading to shared scales inheriting effectively constant relative error from E2M1 regardless of $G$, as visible in Figure 3. **(2) By contrast, for NVFP4, shared scales inherit effectively constant relative error, regardless of $G$**. This is because $S = $ E4M3 is *finer* than $E = $ E2M1, as visible in Figure 3. **(3) Once the Hadamard Transform is applied, the maximum element error is spread across the whole group.** This follows Lemma 1. From Figure 3, one can see that this leads to better precision than pure E2M1, but worse than pure E4M3. Moreover, one can see that for heavy-tailed distribution, such as Laplace or the observed model tensors, $X_{I_\star}^2$ grows faster than $\mathrm{MSE}(G)$ with $G$, leading to the error being *reduced* as we increase the group size $G$. Yet, this effect alone is not enough for it to improve over the E4M3 precision for reasonable group size $G$.

**Discussion.** Our analysis so far showed that the MXFP4 format induces higher MSE for RTN quantization relative to NVFP4, and is worse at outlier preservation. At the same time, the format has lower memory and computational costs relative to NVFP4, and is likely to benefit from normalization via the Hadamard transform. By contrast, the NVFP4 format has *lower MSE* due to the smaller group size, and *top value preservation* as it is "promoted" to E4M3. In addition, the NVFP4 MSE may not benefit from normalizing transforms. In the following, we incorporate our analysis into the classic GPTQ algorithm, obtaining a variant that is designed for FP4 formats, called MR-GPTQ.

## 4 MR-GPTQ: AN FP4-FOCUSED VARIANT OF THE GPTQ ALGORITHM

**Standard GPTQ.** Given a layer's weights $W$ and calibration inputs $X$, GPTQ [21] aims to find quantized weights $\widehat{W}$ that minimize the output reconstruction error: $\|X\widehat{W} - XW\|_2^2$. Assuming a fixed quantization grid, GPTQ builds upon the Optimal Brain Quantization (OBQ) framework [22] to iteratively quantize and update remaining weights to compensate for the error leveraging second-order information, while avoiding OBQ's high computational complexity. Specifically, while OBQ employs a dynamic, greedy weight selection strategy for selecting the next weight to quantize, GPTQ observes that this greedy approach offers low benefits over quantizing weights in an arbitrary, fixed order, for heavily-parameterized layers. Thus, GPTQ quantizes weights across *all rows* in the same fixed order. This enables it to share the Hessian information, used to compute error updates, among rows. GPTQ typically implements this fixed order by processing the dimensions sequentially, column-by-column (front-to-back). The inverse Hessian must be updated only once per column ($d_{\mathrm{col}}$ times) rather than once per weight ($d_{\mathrm{row}} \cdot d_{\mathrm{col}}$ times), which reduces the overall computational complexity from

$O(d_{\text{row}} \cdot d_{\text{col}}^3)$ for OBQ, to $O(\max\{d_{\text{row}} \cdot d_{\text{col}}^2, d_{\text{col}}^3\})$, providing orders-of-magnitude speedup, for a weight matrix of size $d_{\text{row}} \times d_{\text{col}}$.

## 4.1 ADAPTING GPTQ TO FP4 FORMATS

Our analysis showed that, with RTN quantization, NVFP4 provides lower MSE relative to MXFP4, due to better outlier preservation and smaller group size. GPTQ induces an orthogonal direction in the design space, relative to RTN, as it allows for MSE error to be "corrected" by shifting it to other weight blocks. This suggests three general solution strategies: (1) **GPTQ applied to the standard NVFP4 grid**, with `absmax` scaling, leveraging the natural properties of NVFP4. This simply extends RTN to GPTQ; (2) **MR-GPTQ-MXFP4**: GPTQ applied to the MXFP4 grid, on *rotated* weights and activations, as this reduces MSE for RTN; (3) **MR-GPTQ-NVFP4**: GPTQ on an *MSE-optimized* NVFP4 grid, with *rotated* weights and activations.

While the first two approaches follow naturally from our analysis, the third approach wagers that the higher per-group local MSE caused by applying Hadamard rotations to NVFP4 can be compensated by optimizing the scales, together with the GPTQ updates. As such, options 2 and 3 would offer a unified rotated/normalized format, that would apply to both NVFP4 and MXFP4. Next, we describe three key technical additions to the GPTQ algorithm that help bridge the gap between variants.

**Ingredient 1: MSE-Optimized Grids.** Our first step in MR-GPTQ is to identify a good initial grid. Recall that NVFP has both tensor (global) and per-group scales, which we denote by $s_T$ and $s_G$, respectively. The quantized variant of the element $X_i$ will be represented as $\hat{X}_i = s_T \cdot s_G \cdot Q(X_i/(s_T \cdot s_G))$, where $Q$ is the quantization operation. To minimize error, we solve the following optimization problem for each tensor, across its groups: $\min_{s_T, s_{G_1}, \ldots, s_{G_k}} \sum_i [\|\hat{X}_i - X_i\|_2^2$, where $(s_{G_i})_{i=1,k}$ are the quantization scales for the $k$ groups. We solve this by using alternating optimization over the block scales and the per-tensor scale, respectively. For NVFP4 without rotations, we have found this to yield consistent improvements. For MXFP4 with rotations, we have found that a single static value works stably across all layers, and we therefore use this approach in our implementation.

**Ingredient 2: *Static* Activation Reordering.** The original GPTQ algorithm heuristically re-orders the weight columns following the "dynamic act-order", i.e., a descending order of the corresponding Hessian diagonal entries. This matrix shuffle is applied before the quantization grid and scales are computed. While this consistently improves accuracy, it also requires re-shuffling the matrix columns dynamically at runtime, which results in a 10-20% end-to-end inference slow-down.

Instead, we observe that we can apply the activation re-ordering *statically*, i.e. *after* the scales and the quantization grid have been computed in the first step, based on the original (arbitrary) column order. In practice, we first fix the grid and scales for each group, shuffle the columns before GPTQ is applied, and then finally shuffle the columns back, maintaining the microscaling group structure of the original matrix. Importantly, this benefits from the improved behaviour during the quantization process itself, without any runtime penalties. This can be applied to GPTQ over any grid, and provides similar improvements to standard "dynamic" act-order, without the runtime overheads.

**Ingredient 3: Fused Online Rotations.** Our MR-GPTQ variants rotate the weights and activations via a block-wise Hadamard transform $H_k$, with $k \times k$ diagonal blocks, where $k$ is a power-of-two. Mathematically, for a linear layer with weights $W$ and activations $X$, both quantized, the operation occurs as $Q(WH_k)Q(XH_K)^T$, where $H_k$ is the block-wise rotation, and $Q$ is the quantization function. In the next section, we describe how this format can be supported efficiently at runtime.

**MXFP scale fitting** Additionally, for MXFP in Appendix H we propose a simple scale-fitting modification, mapping the excessively large E8M0 range into data range that significantly improves performance of MXFP quantized models.

## 4.2 GPU KERNEL SUPPORT FOR MR-GPTQ VIA QUTLASS

To support the methods described above, we introduce a set of kernels optimized for NVIDIA Blackwell GPUs. These kernels constitute QuTLASS v1.0, a high-performance library for low-precision deep learning quantization, building on NVIDIA CUTLASS [42]. QuTLASS provides full support for quantization- and matmul-related operations in both NVFP4 and MXFP4 micro-scaling formats. In addition, we release architecture-optimized implementations for different NVIDIA

Blackwell compute capabilities, namely SM100 [44] and SM120 [46]. The kernels in QuTLASS can be grouped into two categories, which will handle the computation of $Q(WH_k)Q(XH_k)^T$:

**1. Quantization-related kernels.** While the product $\mathbf{WH_k}$ is pre-fused in the weights, $\mathbf{XH_k}$ occurs online. To avoid diminishing the benefits of FP4 hardware acceleration, QuTLASS provides lightweight fused kernels for online rotation. These kernels support "unimodal" $k \times k$ block diagonal matrices with $k \in \{16, 32, 64, 128\}$. For $k < 256$, dense transformations remain memory-bound, meaning that *any* rotation (not just Hadamards) can be applied at essentially the same cost, as the full matrix can be loaded at runtime (e.g., see Tables 13 and 14). To further reduce overhead, quantization and scale calculation are fused into the transformation kernel as a custom epilogue function. QuTLASS currently supports MSE and Abs-Max quantization methods, while its template-based design allows new methods to be easily integrated.

**2. Matmul-related narrow precision kernels.** Between FP4 quantization and matrix multiplication, a hardware-mandated rearrangement of scaling factors is required [45] for `tcgen05.mma`. QuT-LASS implements this step using a Triton kernel. For the matmul itself, QuTLASS supports multiple backends, including CUTLASS [42] and FlashInfer [57], enabling flexible plug-and-play backend selection depending on workload and hardware.

## 5 EXPERIMENTAL RESULTS

**1. Experiments with Emulated Quantization.** We first evaluate the highly-popular Llama 3.1-8B-Instruct model [18], examining the impact of quantizing both weights and activations for all linear layers in this model to the INT4 and FP4 formats, using different algorithms. To ensure compatibility, experiments are performed using simulated quantization in PyTorch. We use a subset of tasks from the Open LLM Leaderboard V1 [5] for evaluation: GSM8K for grade school math [10], MMLU for world knowledge and reasoning [26; 9], Winogrande and HellaSwag for language understanding [48; 58]. (Other tasks in this harness yield similar scores across top methods.) The INT4 experiments use group size 32 with FP16 scales, matching the average bit-width of NVFP4.

**Algorithms.** We consider both weights-and-activations quantization (W4A4, our main focus) and weight-only quantization (W4A16, as a "control"). For W4A4, we implement the following: (1) **Round-to-nearest (RTN)** quantization to the corresponding format, with `absmax` scales. In addition, we add Hadamard rotations matching the quantization group size (32), denoted as **RTN + HT**. (2) **SmoothQuant** [54] diagonal rescaling, with a tuned $\alpha$ smoothening factor. We identified $\alpha = 0.6$ to be the best in our experiments. (3) **QuaRot** [4], which adds Hadamard rotations strategically at each linear layer. These should reduce quantization error, and most of them can be folded into the model. We use RTN for quantization post-rotation. (4) **SpinQuant** [38], which adds trainable rotations to the model, similarly to QuaRot. A subset of 1024 calibration sequences from FineWeb is used for training the matrices. (5) **GPTQ** [21] weight quantization and RTN on the activations, with `absmax` scales. A subset of 1024 calibration sequences from FineWeb, `absmax` scales, standard Hessian dampening factors ($\lambda = 10^{-2}$), and standard quantization order are used. (6) **MR-GPTQ** weight quantization, i.e., GPTQ with block rotations, MSE scale optimization, and static activation re-ordering over the rotated weights, as described in Section 4.1, with RTN on the activations. As a control, we also implement **weight-only quantization**, via **RTN**, **GPTQ**, **AWQ** [36], as well as Hadamard rotations followed by RTN, denoted as **RTN + HT**. These results closely follow our findings for W&A quantization, and are thus deferred to the Appendix. In Appendix K, we perform an exhaustive sweep over DCT, DST, Hadamard, and GSR transforms and block sizes showing that the Hadamard transform matching the quantization group size provides the best results on average.

**Discussion.** The accuracy results for W4A4 experiments on Llama-3.1-8B-Instruct are presented in Table 1. The variance for the NVFP4 experiments (i.e., for entries in the 7th column over 5 distinct seeds) is of approximately 0.3 average points, whereas the variance for the INT4 experiments is of approximately 1 point. We mark all top aggregate entries (within 2 standard deviations) as bold in the corresponding columns. We observe the following:

**(1) No Lossless Format**: Across all formats, the accuracy drop is noticeable. The lowest average drop is for the NVFP4 format with SmoothQuant, GPTQ, or MR-GPTQ (these results are within variance of each other). The weight quantization results (Appendix Table 2), show that the induced error is roughly evenly split between weight and activation quantization. These results suggest that micro-scaling is not a direct solution for accuracy recovery. **(2) NVFP4 provides the best accuracy,**

| Format | Method | MMLU-CoT | GSM8k | HellaSwag | WinoGrande | Avg. | Recovery % |
|---|---|---|---|---|---|---|---|
| Baseline | FP16 | 72.76 | 85.06 | 80.01 | 77.90 | **78.93** | **100** |
| INT8 | RTN | 72.50 | 84.80 | 80.20 | 77.40 | **78.73** | **99.74** |
| | GPTQ | 72.40 | 84.40 | 80.00 | 77.30 | **78.53** | **99.48** |
| FP8 | RTN | 72.40 | 84.70 | 79.80 | 77.70 | **78.65** | **99.64** |
| | GPTQ | 71.80 | 84.50 | 79.90 | 78.10 | **78.58** | **99.55** |
| INT4 | RTN | 65.96 | 74.68 | 77.62 | 74.19 | 73.11 | 92.63 |
| | RTN+HT | 68.30 | 79.61 | 77.60 | 73.48 | **74.75** | **94.71** |
| | GPTQ | 66.36 | 76.65 | 77.38 | 72.48 | 73.21 | 92.75 |
| NVFP4 | RTN | 68.26 | 78.39 | 78.15 | 74.11 | 74.73 | 94.67 |
| | RTN + HT | 67.41 | 78.01 | 77.31 | 73.48 | 74.05 | 93.82 |
| | QuaRot | 66.50 | 77.40 | 77.25 | 75.14 | 74.10 | 93.80 |
| | SpinQuant | 66.50 | 76.10 | 76.96 | 75.32 | 73.70 | 93.40 |
| | SmoothQuant | 68.90 | 79.50 | 79.50 | 74.70 | **75.70** | **95.90** |
| | GPTQ | 68.85 | 82.60 | 78.26 | 74.51 | **75.72** | **95.92** |
| | MR-GPTQ | 69.12 | 80.80 | 78.17 | 75.24 | **75.84** | **96.08** |
| MXFP4 | RTN | 62.21 | 67.85 | 73.99 | 73.24 | 69.32 | 87.83 |
| | RTN + HT | 62.38 | 72.48 | 75.29 | 71.67 | 70.45 | 89.26 |
| | SmoothQuant | 63.93 | 68.54 | 75.10 | 73.56 | 70.30 | 89.06 |
| | QuaRot | 49.86 | 56.94 | 73.50 | 71.43 | 62.90 | 79.70 |
| | SpinQuant | 61.80 | 68.16 | 74.87 | 72.93 | 69.40 | 88.00 |
| | GPTQ | 63.49 | 68.46 | 76.01 | 74.51 | 70.62 | 89.47 |
| | MR-GPTQ | 67.19 | 75.70 | 76.91 | 74.80 | **73.65** | **93.31** |
| NVINT4 | RTN | 68.56 | 78.17 | 78.64 | 75.14 | 75.13 | 95.18 |
| | RTN + HT | 68.59 | 81.73 | 78.38 | 74.27 | 75.74 | 95.96 |
| | GPTQ | 68.69 | 81.58 | 77.59 | 73.40 | 75.32 | 95.42 |
| | MR-GPTQ | 69.71 | 82.26 | 79.14 | 75.53 | **76.66** | **97.12** |
| MXINT4 | RTN | 55.06 | 56.79 | 72.06 | 68.27 | 63.05 | 79.87 |
| | RTN + HT | 58.44 | 61.64 | 73.94 | 71.19 | 66.30 | 84.00 |
| | GPTQ | 61.22 | 67.70 | 75.04 | 71.67 | 68.91 | 87.30 |
| | MR-GPTQ | 65.48 | 74.83 | 76.63 | 73.09 | **72.51** | **91.86** |

Table 1: Unified accuracy comparison of Llama-3.1-8B-Instruct W4A4 under different quantization formats and methods. For each format, top methods within variance are marked in bold.

**with INT4 second, and MXFP4 third**: On average, NVFP4 and INT4 quantization provide similar quality, with INT4 quantization having higher variance. The MXFP4 format is a distant third in terms of accuracy, regardless of the method used, but benefits significantly from MR-GPTQ. **(3) Quantization Method Efficiency:** First, we note the good performance of standard RTN for INT4 (with rotations) and NVFP4 (without). Second, the Hadamard transform appears effective for INT4 and MXFP4 (which use group size 32), but is less effective for NVFP4 (which uses group size 16), confirming our analysis. In particular, for round-to-nearest quantization, adding the Hadamard transform to NVFP4 *hurts* accuracy. Finally, the GPTQ and SmoothQuant methods appear to be consistently—but moderately—effective across all three formats.

**2. Real Quantization.** We integrate our kernels in vLLM [30], and perform accuracy evaluations directly in this setup over additional models, such as Llama-3.3-70B-Instruct [18], and the Qwen3 [55] family of models. The results are presented in Table 10 and Figure 7. For this experiment, we also provide results for Quantization-Aware Training (QAT) performed using the balanced Generalized Jensen-Shannon Divergence loss [20; 33] between the quantized and the unquantized (frozen) model token distributions on a subset of 92,995 samples (10%) from the Tülu 3 [31] instructions dataset. The results show that accuracies measured over real kernels for the Llama-3.1-8B-Instruct model track closely with the results from simulation, with slightly lower recoveries (within 0.2-0.3%). Smaller models (< 8B) and Llama-family models tend to have lower recovery rates, whereas Qwen3 models can achieve more than 99% average recovery in NVFP4. For NVFP4, standard GPTQ provides the highest recoveries on average, although RTN and MR-GPTQ are also competitive, with QAT only providing very limited benefits. For MXFP4, MR-GPTQ provides the best recovery among PTQ methods, while QAT consistently reduces the gap to full precision. Additionally, in Appendix I we analyze differences between GPTQ variants on the less noisy PlatinumBench benchmark [53].

**3. Integer Microscaling Formats.** Although integer-based microscaling formats are not specified in the OCP standard [47], one could easy extend the idea to them. For validation purposes, we propose the two following microscaling INT4-based formats: *NVINT4* and *MXINT4* that are identical to NVFP4 and MXFP4 respectively except for the base type being INT4 instead of FP4. The error and outlier preservation analysis in Figure 10 predicts that NVINT4 should profit from HT and yield performance superior to all other formats. Pseudo-quantization evaluations in Tables 1, 7 and Figure 7 verify that for 0-shot (RTN) and 1-shot (MR-GPTQ) quantization — a finding similar to that of Chen et al. [7].

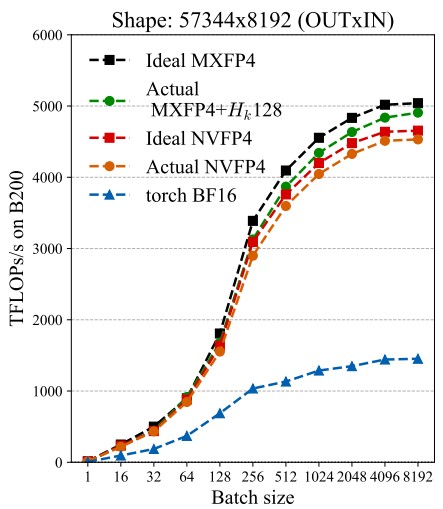

Figure 5: QuTLASS performance for a single linear LLM layer.

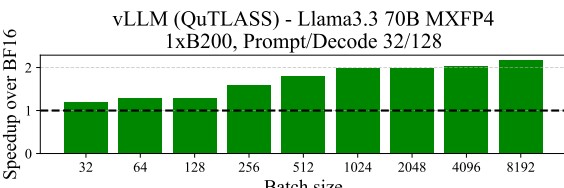

Figure 6: QuTLASS performance end-to-end using our vLLM integration.

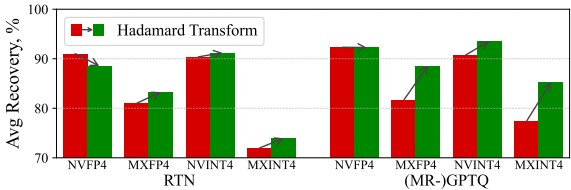

Figure 7: Average (over Llama 3 1,3,8B) recoveries for microscaling formats and the effect of HT.

**4. Kernel and Inference Performance.** Finally, in Figures 5 and 6, we examine the performance of our kernels. Figure 5 shows throughput for a single layer extracted from a Llama-3.3-70B model using FlashInfer as a backend. The curve labeled with "ideal" represents the upper bound for a real 4-bit weight and 4-bit activation matrix multiplication, i.e., the measured matmul throughput not including the overhead of quantization-related operations. In contrast, the curves labeled "actual" show real measurements including the costs of Hadamards, quantization, and scale computation. The comparison highlights the small gap between idealized efficiency and practical implementations with our kernels, with speedups of up to $\approx 3.6\times$ (out of $4\times$) on B200 and $\approx 6\times$ (out of $8\times$) on RTX5090. Interestingly, MXFP4 *outperforms* NVFP4 on B200, with up to $\approx 15\%$ higher throughput, despite their closely related numerical formats. Possible contributing factors include MXFP4's use of potentially more efficient power-of-two scales as well as larger group sizes, which could reduce overhead.

Figure 6 shows end-to-end speedup of vLLM running Llama-3.3-70B with MXFP4 quantization compared to the baseline BF16 implementation on a single B200 GPU. The results demonstrate consistent performance gains across batch sizes, with speedups reaching up to $2.2\times$ over the BF16 baseline, and nearly $4\times$ on an RTX 5090 GPU (see Appendix F for more details).

# 6 CONCLUSION

We present a first comprehensive study of the recently introduced MXFP4 and NVFP4 formats for LLM quantization, revealing gaps between the promise of these formats and their performance using state-of-the-art methods. To bridge these gaps, we introduce Micro-Rotated-GPTQ (MR-GPTQ), a novel GPTQ variant adapted to these formats. We support this approach with QuTLASS, a suite of high-performance GPU kernels that implement MR-GPTQ's micro-rotations with negligible overhead. We hope that our results will provide a basis and a motivation for future work on improving accuracy for these novel formats.

## REPRODUCIBILITY STATEMENT

**Code.** The source code required to reproduce the quantization and accuracy-evaluation experiments is available in the FP-Quant repository. The implementation of QuTLASS can be accessed here.

**Models.** MR-GPTQ quantized models are hosted in a dedicated Hugging Face collection.

## ACKNOWLEDGMENTS

We thank members of the Red Hat AI team, particularly Rob Shaw, Michael Goin, and Kyle Sayers, for their support and useful suggestions. We would like to thank Lambda Cloud for their generous computational grant. We thank the NVIDIA and Google corporation for their grants, which supported part of this research. Roberto L. Castro and Andrei Panferov were supported in part by the BILAI Cluster of Excellence program (GreenAI), whereas Vage Egiazarian and Eldar Kurtic were supported by the ERC Proof-of-Concept Grant FastAI. This work was supported under project ID 40 as part of the Swiss AI Initiative, through a grant from the ETH Domain and computational resources provided by the Swiss National Supercomputing Centre (CSCS) under the Alps infrastructure.

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

## TECHNICAL APPENDICES AND SUPPLEMENTARY MATERIAL

## A WEIGHT-ONLY QUANTIZATION RESULTS

The results for weight-only quantization are provided in Table 2. One can observe that similary to the weight and activation quantization case INT4 and NVFP4 perform similarly, while MXFP suffers much significant accuracy drop. Even for weight-only case there is 2% accuracy drop on average relative to the original model.

| Format | Quantization | MMLU | GSM8k | HellaSwag | WinoGrande | Avg. | Recovery% |
|--------|--------------|------|-------|-----------|------------|------|-----------|
| FP16 | - | 72.80 | 85.10 | 80.00 | 78.90 | 79.70 | – |
| INT4 | RTN | 69.38 | 81.80 | 79.41 | 77.90 | **77.12** | **97.71** |
| | RTN+Had | 70.27 | 82.56 | 79.18 | 76.64 | **77.16** | **97.76** |
| | GPTQ | 70.25 | 80.52 | 79.01 | 76.64 | **76.60** | **97.05** |
| NVFP4 | RTN | 70.64 | 82.26 | 79.24 | 77.35 | **77.37** | **98.02** |
| | RTN+Had | 69.26 | 80.82 | 78.52 | 77.03 | 76.41 | 96.80 |
| | GPTQ | 70.52 | 82.49 | 79.35 | 76.95 | **77.33** | **97.96** |
| | AWQ | 70.57 | 82.71 | 79.30 | 77.03 | **77.40** | **98.06** |
| MXFP | RTN | 68.23 | 80.36 | 77.26 | 75.93 | 75.44 | 95.58 |
| | RTN+Had | 66.24 | 77.56 | 77.34 | 74.11 | 73.81 | 93.51 |
| | GPTQ | 68.79 | 81.43 | 78.40 | 76.88 | **76.37** | **96.76** |
| | AWQ | 68.16 | 78.70 | 78.56 | 75.30 | 75.18 | 95.25 |

Table 2: Performance of Llama-3.1-8B-Instruct under different weight-only quantization settings.

## B REAL QUANTIZATION RESULTS

In this section we provide a complete set of evaluation results for Llama-3 (Llama-3.2-1B-Instruct, Llama-3.2-3B-Instruct, Llama-3.1-8B-Instruct, Llama-3.3-70B-Instruct) and Qwen-3 (Qwen-3-8B, Qwen-3-14B, Qwen-3-32B) model families. We turn off thinking mode for Qwen as it turned out that long reasoning chains-of-thought turned out to be detrimental for performance on GSM8k and MMLU-CoT. The scores were produced using QuTLASS vLLM integration.

Table 10 present more details on accuracy recovery for micro-scaling formats, including *NVINT4* and *MXINT4* and QAT.

| Format | Quantization | MMLU | GSM8k | HellaSwag | WinoGrande | Avg. | Recovery% |
|---|---|---|---|---|---|---|---|
| - | FP16 | 46.20 | 46.32 | 59.78 | 61.56 | 53.47 | – |
| INT8 | RTN | 45.90 | 44.20 | 59.80 | 61.30 | **52.80** | **99.55** |
| | GPTQ | 45.40 | 44.90 | 59.60 | 60.10 | **52.50** | **98.99** |
| FP8 | RTN | 46.10 | 44.70 | 59.50 | 59.50 | **52.50** | **98.99** |
| | GPTQ | 45.80 | 45.00 | 59.10 | 60.60 | **52.63** | **99.22** |
| NVFP | RTN | 36.08 | 31.39 | 54.77 | 57.22 | 44.87 | 83.91 |
| | RTN+Had16 | 32.80 | 25.02 | 56.24 | 59.04 | 43.28 | 80.94 |
| | RTN+Had128 | 38.28 | 29.95 | 54.27 | 58.41 | 45.23 | 84.59 |
| | GPTQ | 37.79 | 29.80 | 55.48 | 60.22 | 45.82 | 85.71 |
| | GPTQ+Had16 | 38.99 | 32.98 | 56.66 | 58.17 | **46.70** | **87.35** |
| | GPTQ+Had128 | 35.47 | 31.16 | 57.02 | 59.19 | 45.71 | 85.50 |
| | QAT | 27.85 | 38.51 | 57.52 | 60.30 | 46.05 | 86.12 |
| | QAT+Had16 | 32.72 | 37.60 | 57.53 | 58.41 | **46.57** | **87.09** |
| MXFP | RTN | 30.46 | 11.83 | 48.28 | 54.22 | 36.20 | 67.70 |
| | RTN+Had32 | 30.89 | 19.41 | 51.64 | 57.22 | 39.79 | 74.42 |
| | RTN+Had128 | 34.48 | 25.55 | 53.98 | 58.01 | 43.01 | 80.44 |
| | GPTQ | 26.84 | 13.50 | 49.29 | 56.75 | 36.60 | 68.45 |
| | GPTQ+Had32 | 29.44 | 27.60 | 54.89 | 58.72 | 42.66 | 79.80 |
| | GPTQ+Had128 | 35.68 | 28.13 | 54.60 | 58.72 | 44.28 | 82.83 |
| | QAT | 15.60 | 20.32 | 53.34 | 56.51 | 36.44 | 68.16 |
| | QAT+Had32 | 28.12 | 36.85 | 57.04 | 58.80 | **45.20** | **84.55** |
| NVINT4 | RTN | 37.33 | 26.08 | 52.62 | 58.56 | 43.65 | 81.64 |
| | RTN+Had16 | 33.41 | 32.52 | 57.12 | 59.19 | 45.56 | 85.21 |
| | GPTQ | 37.15 | 27.60 | 55.94 | 59.35 | 45.01 | 84.19 |
| | MR-GPTQ | 36.69 | 33.36 | 57.95 | 58.96 | **46.74** | **87.42** |
| MXINT4 | RTN | 21.85 | 4.55 | 45.07 | 55.49 | 31.74 | 59.37 |
| | RTN+Had32 | 13.17 | 9.48 | 48.91 | 53.99 | 31.39 | 58.71 |
| | GPTQ | 23.42 | 13.27 | 50.02 | 55.88 | 35.65 | 66.67 |
| | MR-GPTQ | 21.81 | 23.12 | 54.96 | 55.41 | **38.83** | **72.62** |

Table 3: Performance of Llama-3.2-1B-Instruct for different weight & activation quantization settings.

| Format | Quantization | MMLU | GSM8k | HellaSwag | WinoGrande | Avg. | Recovery% |
|---|---|---|---|---|---|---|---|
| - | FP16 | 64.43 | 78.01 | 73.42 | 70.09 | 71.49 | – |
| INT8 | RTN | 64.00 | 77.70 | 73.30 | 69.60 | **71.15** | **99.47** |
| | GPTQ | 64.00 | 77.90 | 73.40 | 69.90 | **71.30** | **99.67** |
| FP8 | RTN | 63.40 | 77.70 | 73.00 | 69.70 | **70.95** | **99.19** |
| | GPTQ | 64.10 | 77.80 | 73.00 | 69.90 | **71.20** | **99.54** |
| NVFP | RTN | 60.62 | 70.43 | 70.99 | 68.03 | 67.52 | 94.45 |
| | RTN+Had16 | 59.91 | 64.82 | 69.77 | 65.59 | 65.02 | 90.96 |
| | RTN+Had128 | 54.34 | 67.48 | 69.69 | 66.93 | 64.61 | 90.38 |
| | GPTQ | 61.76 | 70.36 | 71.07 | 69.93 | 68.28 | 95.51 |
| | GPTQ+Had16 | 60.26 | 68.76 | 71.05 | 67.80 | 66.97 | 93.68 |
| | GPTQ+Had128 | 60.19 | 70.89 | 70.97 | 68.19 | 67.56 | 94.51 |
| | MicroQAT+Had16 | 60.66 | 69.98 | 70.55 | 67.01 | 67.05 | 93.79 |
| | QAT | 62.06 | 75.06 | 71.27 | 67.96 | **69.09** | **96.64** |
| | QAT+Had16 | 62.03 | 72.93 | 70.95 | 66.46 | 68.09 | 95.25 |
| MXFP | RTN | 56.81 | 60.80 | 67.30 | 64.56 | 62.37 | 87.24 |
| | RTN+Had32 | 55.58 | 57.77 | 68.56 | 64.33 | 61.56 | 86.11 |
| | RTN+Had128 | 55.95 | 60.80 | 67.57 | 64.88 | 62.30 | 87.15 |
| | GPTQ | 57.68 | 62.32 | 63.87 | 64.88 | 62.19 | 86.99 |
| | GPTQ+Had32 | 59.79 | 68.92 | 69.50 | 66.85 | 66.27 | 92.69 |
| | GPTQ+Had128 | 59.56 | 67.78 | 70.08 | 68.03 | 66.36 | 92.83 |
| | MicroQAT+Had32 | 59.49 | 65.66 | 69.05 | 67.32 | 65.38 | 91.46 |
| | QAT | 56.17 | 64.90 | 69.51 | 67.17 | 64.44 | 90.14 |
| | QAT+Had32 | 59.83 | 72.48 | 70.27 | 66.54 | **67.28** | **94.11** |
| NVINT4 | RTN | 60.22 | 71.65 | 70.92 | 66.77 | 67.39 | 94.27 |
| | RTN + HT | 56.75 | 71.95 | 69.24 | 66.61 | 66.14 | 92.52 |
| | GPTQ | 60.5 | 71.42 | 27.01 | 51.07 | 52.50 | 73.44 |
| | MR-GPTQ | 60.8 | 72.25 | 71.74 | 70.48 | **68.82** | **96.27** |
| MXINT4 | RTN | 46.03 | 47.54 | 64.21 | 61.56 | 54.84 | 76.71 |
| | RTN + HT | 50.55 | 51.33 | 64.61 | 59.83 | 56.58 | 79.15 |
| | GPTQ | 53.48 | 60.73 | 51.44 | 58.80 | 56.11 | 78.49 |
| | MR-GPTQ | 57.73 | 67.48 | 68.92 | 66.69 | **65.21** | **91.21** |

Table 4: Performance of Llama-3.2-3B-Instruct for different weight & activation quantization settings.

| Format | Quantization | MMLU-CoT | GSM8k | HellaSwag | WinoGrande | Avg. | Recovery% |
|---|---|---|---|---|---|---|---|
| - | FP16 | 72.80 | 85.10 | 80.00 | 77.90 | 78.90 | – |
| INT8 | RTN | 72.50 | 84.80 | 80.20 | 77.40 | **78.73** | **99.74** |
| | GPTQ | 72.40 | 84.40 | 80.00 | 77.30 | **78.53** | **99.48** |
| FP8 | RTN | 72.40 | 84.70 | 79.80 | 77.70 | **78.65** | **99.64** |
| | GPTQ | 71.80 | 84.50 | 79.90 | 78.10 | **78.58** | **99.55** |
| NVFP | RTN | 68.70 | 78.70 | 78.40 | 73.40 | 74.80 | 94.80 |
| | RTN+Had | 67.00 | 77.40 | 77.30 | 74.40 | 74.00 | 93.80 |
| | RTN+Had128 | 66.60 | 77.00 | 77.50 | 75.50 | 74.10 | 93.90 |
| | GPTQ | 68.60 | 79.60 | 78.70 | 75.50 | 75.60 | 95.70 |
| | GPTQ+Had | 69.40 | 79.60 | 78.40 | 75.10 | 75.60 | 95.80 |
| | GPTQ+Had128 | 68.90 | 79.50 | 78.30 | 73.60 | 75.10 | 95.10 |
| | QAT | 68.20 | 79.80 | 78.90 | 74.40 | 75.30 | 95.40 |
| | QAT+Had | 68.90 | 81.60 | 79.00 | 75.10 | **76.10** | **96.50** |
| MXFP | RTN | 62.20 | 69.50 | 73.80 | 72.60 | 69.50 | 88.10 |
| | RTN+Had | 62.60 | 71.80 | 75.20 | 72.30 | 70.50 | 89.30 |
| | RTN+Had128 | 64.50 | 72.70 | 76.00 | 73.30 | 71.60 | 90.70 |
| | GPTQ | 63.74 | 70.20 | 75.52 | 7364 | 70.78 | 89.66 |
| | GPTQ+Had | 67.20 | 77.50 | 77.00 | 73.10 | 73.70 | 93.30 |
| | GPTQ+Had128 | 66.80 | 78.30 | 76.90 | 74.90 | 74.20 | 94.00 |
| | QAT | 65.00 | 76.00 | 77.60 | 72.90 | 72.90 | 92.30 |
| | QAT+Had | 67.60 | 80.30 | 78.30 | 74.90 | **75.30** | **95.40** |
| NVINT4 | RTN | 68.56 | 78.17 | 78.64 | 75.14 | 75.13 | 95.18 |
| | RTN + HT | 68.59 | 81.73 | 78.38 | 74.27 | 75.74 | 95.96 |
| | GPTQ | 68.69 | 81.58 | 77.59 | 73.40 | 75.32 | 95.42 |
| | MR-GPTQ | 69.71 | 82.26 | 79.14 | 75.53 | **76.66** | **97.12** |
| MXINT4 | RTN | 55.06 | 56.79 | 72.06 | 68.27 | 63.05 | 79.87 |
| | RTN + HT | 58.44 | 61.64 | 73.94 | 71.19 | 66.30 | 84.00 |
| | GPTQ | 61.22 | 67.70 | 75.04 | 71.67 | 68.91 | 87.30 |
| | MR-GPTQ | 65.48 | 74.83 | 76.63 | 73.09 | **72.51** | **91.86** |

Table 5: Performance of Llama-3.1-8B-Instruct for different weight & activation quantization settings.

| Format | Quantization | MMLU | GSM8k | HellaSwag | WinoGrande | Avg. | Recovery% |
|---|---|---|---|---|---|---|---|
| - | FP16 | 86.55 | 95.07 | 86.22 | 84.93 | 88.19 | – |
| NVFP | RTN | 85.50 | 93.48 | 85.63 | 83.27 | 86.97 | 98.61 |
| | RTN+Had16 | 85.02 | 93.63 | 84.97 | 83.82 | 86.86 | 98.49 |
| | RTN+Had128 | 85.24 | 91.81 | 84.91 | 83.35 | 86.33 | 97.89 |
| | GPTQ | 85.54 | 94.09 | 85.49 | 84.37 | 87.37 | 99.07 |
| | GPTQ+Had16 | 85.58 | 93.40 | 85.45 | 82.40 | 86.71 | 98.32 |
| | GPTQ+Had128 | 85.59 | 94.16 | 85.56 | 84.77 | **87.52** | **99.24** |
| MXFP | RTN | 83.42 | 92.65 | 83.93 | 81.45 | 85.36 | 96.79 |
| | RTN+Had32 | 83.86 | 93.56 | 84.13 | 83.58 | 86.28 | 97.83 |
| | RTN+Had128 | 84.37 | 94.47 | 84.22 | 82.40 | 86.37 | 97.93 |
| | GPTQ | 83.77 | 94.47 | 84.41 | 82.64 | 86.32 | 97.88 |
| | GPTQ+Had32 | 84.82 | 94.54 | 84.66 | 83.11 | 86.78 | 98.40 |
| | GPTQ+Had128 | 84.90 | 93.90 | 84.80 | 83.80 | **86.86** | **98.48** |

Table 6: Performance of Llama-3.3-70B-Instruct for different weight & activation quantization settings.

| Format | Quantization | MMLU | GSM8k | HellaSwag | WinoGrande | Avg. | Recovery% |
|---|---|---|---|---|---|---|---|
| - | FP16 | 72.98 | 90.90 | 75.52 | 70.56 | 77.49 | – |
| NVFP | RTN | 70.78 | 90.30 | 74.63 | 70.72 | **76.61** | **98.86** |
| | RTN+Had16 | 70.19 | 86.35 | 73.02 | 68.11 | 74.42 | 96.04 |
| | RTN+Had128 | 69.09 | 86.66 | 73.47 | 67.96 | 74.30 | 95.88 |
| | GPTQ | 70.90 | 88.17 | 75.01 | 70.09 | 76.04 | 98.13 |
| | GPTQ+Had16 | 71.06 | 88.32 | 74.58 | 68.03 | 75.50 | 97.43 |
| | GPTQ+Had128 | 70.45 | 87.41 | 74.25 | 68.90 | 75.25 | 97.11 |
| | QAT | 70.94 | 89.08 | 74.67 | 68.51 | 75.80 | 97.82 |
| | QAT+Had16 | 71.34 | 89.23 | 75.24 | 70.40 | **76.55** | **98.79** |
| MXFP | RTN | 67.69 | 84.23 | 71.24 | 67.40 | 72.64 | 93.74 |
| | RTN+Had32 | 67.57 | 83.78 | 71.32 | 67.32 | 72.50 | 93.56 |
| | RTN+Had128 | 67.27 | 81.58 | 71.41 | 66.38 | 71.66 | 92.48 |
| | GPTQ | 68.01 | 84.23 | 71.65 | 67.80 | 72.92 | 94.11 |
| | GPTQ+Had32 | 69.13 | 84.84 | 73.17 | 68.03 | 73.79 | 95.23 |
| | GPTQ+Had128 | 69.53 | 86.43 | 73.55 | 65.75 | 73.82 | 95.26 |
| | QAT | 69.45 | 87.34 | 74.03 | 69.85 | 75.17 | 97.00 |
| | QAT+Had32 | 70.35 | 89.61 | 74.61 | 70.56 | **76.28** | **98.44** |

Table 7: Performance of Qwen-8B for different weight & activation quantization settings.

| Format | Quantization | MMLU | GSM8k | HellaSwag | WinoGrande | Avg. | Recovery% |
|---|---|---|---|---|---|---|---|
| - | FP16 | 77.18 | 91.96 | 79.84 | 74.27 | 80.81 | – |
| NVFP | RTN | 75.73 | 91.28 | 78.36 | 73.16 | 79.63 | 98.54 |
| | RTN+Had16 | 74.98 | 92.04 | 77.76 | 72.38 | 79.29 | 98.12 |
| | RTN+Had128 | 74.46 | 91.13 | 77.60 | 71.98 | 78.79 | 97.50 |
| | GPTQ | 74.88 | 91.28 | 78.40 | 74.51 | 79.77 | 98.71 |
| | GPTQ+Had16 | 75.49 | 91.43 | 78.38 | 74.51 | **79.95** | **98.94** |
| | GPTQ+Had128 | 75.10 | 90.52 | 78.30 | 72.77 | 79.17 | 97.97 |
| MXFP | RTN | 72.92 | 90.22 | 76.68 | 71.51 | 77.83 | 96.31 |
| | RTN+Had32 | 73.19 | 89.54 | 75.95 | 71.67 | 77.59 | 96.01 |
| | RTN+Had128 | 73.17 | 85.60 | 76.80 | 72.14 | 76.93 | 95.19 |
| | GPTQ | 72.57 | 89.54 | 76.50 | 72.45 | 77.77 | 96.23 |
| | GPTQ+Had32 | 74.36 | 89.92 | 77.64 | 72.53 | **78.61** | **97.28** |
| | GPTQ+Had128 | 74.11 | 89.92 | 77.77 | 71.11 | 78.23 | 96.80 |

Table 8: Performance of Qwen-14B for different weight & activation quantization settings.

| Format | Quantization | MMLU | GSM8k | HellaSwag | WinoGrande | Avg. | Recovery% |
|---|---|---|---|---|---|---|---|
| - | FP16 | 80.81 | 92.04 | 83.97 | 76.56 | 83.35 | – |
| NVFP | RTN | 79.85 | 94.24 | 83.27 | 75.22 | **83.15** | **99.76** |
| | RTN+Had16 | 78.90 | 89.23 | 82.60 | 76.48 | 81.80 | 98.15 |
| | RTN+Had128 | 78.49 | 89.69 | 82.47 | 75.37 | 81.51 | 97.79 |
| | GPTQ | 79.54 | 92.87 | 83.24 | 75.93 | **82.90** | **99.46** |
| | GPTQ+Had16 | 78.60 | 90.90 | 82.93 | 75.14 | 81.89 | 98.26 |
| | GPTQ+Had128 | 79.11 | 90.52 | 83.15 | 76.09 | 82.22 | 98.65 |
| MXFP | RTN | 77.07 | 72.33 | 81.52 | 75.22 | 76.54 | 91.83 |
| | RTN+Had32 | 78.22 | 93.03 | 81.76 | 75.93 | **82.24** | **98.67** |
| | RTN+Had128 | 78.36 | 88.10 | 81.66 | 75.30 | 80.86 | 97.01 |
| | GPTQ | 77.01 | 88.55 | 81.79 | 74.90 | 80.56 | 96.66 |
| | GPTQ+Had32 | 78.46 | 82.41 | 82.72 | 75.06 | 79.66 | 95.58 |
| | GPTQ+Had128 | 78.90 | 90.90 | 82.29 | 75.22 | **81.83** | **98.18** |

Table 9: Performance of Qwen-32B for different weight & activation quantization settings.

Table 10: Per-model recoveries with real (NVFP4 and MXFP4) and hypothetical (NVINT4 and MXINT4) quantization.

| Format | Method | HT | Llama3 | | | | Qwen3 | | |
|---|---|---|---|---|---|---|---|---|---|
| | | | 1B | 3B | 8B | 70B | 8B | 14B | 32B |
| NVFP4 | RTN | – | 83.9 | 94.4 | 94.8 | 98.6 | 98.9 | 98.5 | 99.8 |
| | RTN | 16 | 80.9 | 91.0 | 93.8 | 98.5 | 96.0 | 98.1 | 98.1 |
| | GPTQ | – | 85.7 | 95.5 | 95.7 | 99.1 | 98.1 | 98.7 | 99.5 |
| | MR-GPTQ | 16 | 87.3 | 93.7 | 95.8 | 98.3 | 97.4 | 98.9 | 98.3 |
| | QAT | – | 86.1 | 96.6 | 95.4 | – | 97.8 | – | – |
| | QAT | 16 | 87.1 | 95.3 | 96.5 | – | 98.8 | – | – |
| MXFP4 | RTN | – | 67.7 | 87.2 | 88.1 | 96.8 | 93.7 | 96.3 | 91.8 |
| | RTN | 32 | 74.4 | 86.1 | 89.3 | 97.8 | 93.6 | 96.0 | 98.7 |
| | GPTQ | – | 68.4 | 87.0 | 89.7 | 97.9 | 94.1 | 96.2 | 96.7 |
| | MR-GPTQ | 32 | 79.8 | 92.7 | 93.3 | 98.4 | 95.2 | 97.3 | 95.6 |
| | QAT | – | 68.2 | 90.1 | 92.3 | – | 97.0 | – | – |
| | QAT | 32 | 84.5 | 94.1 | 95.4 | – | 98.4 | – | – |
| NVINT4 | RTN | – | 81.6 | 94.3 | 95.2 | – | 96.3 | 98.1 | – |
| | RTN | 16 | 85.2 | 92.5 | 96.0 | – | 98.1 | 99.2 | – |
| | GPTQ | – | 84.2 | 92.5 | 95.4 | – | – | – | – |
| | MR-GPTQ | 16 | 87.4 | 96.3 | 97.1 | – | – | – | – |
| MXINT4 | RTN | – | 59.4 | 76.7 | 79.9 | – | 83.9 | 92.4 | – |
| | RTN | 32 | 58.7 | 79.1 | 84.0 | – | 89.7 | 94.9 | – |
| | GPTQ | – | 66.7 | 78.5 | 87.3 | – | – | – | – |
| | MR-GPTQ | 32 | 72.6 | 91.2 | 91.9 | – | – | – | – |

## C  SCALE QUANTIZATION ANALYSIS

As discussed in the main text, microscaling formats adopt scale quantization to reduce memory storage overhead and accelerate dequantization operations. However, scale quantization may introduce additional error due to rounding of scales onto a coarser grid. Below we provide an analysis and explore alternative choices for scale quantization.

MXFP format adopts E8M0 grid with exponentially spaced levels. It allows to represent values with very small and large magnitude, yet the distance between adjacent levels can be pretty large resulting in large approximation errors. E4M3 grid used in NVFP, on the other hand, has much narrower dynamic range $[-448, 448]$ with levels spread more uniformly. We note, that the sign bit is in fact never utilized, given that the scale is a non-negative value by definition.

Below, we explore several choices for 8-bit scale quantization with a fixed group size of 16. Specifically, we measure weight and activation $\text{MSE}^{\text{rel}}$ for a range of EeMm formats with e + m = 7, as well as for E8M0 and INT8. For E8M0 scale quantization, we multiply the scale by 4/3 following [52], which yields an unbiased estimate of the original scale and reduces quantization error. Results for weight and activation quantization are shown in Figure 8 and Figure 9, respectively.

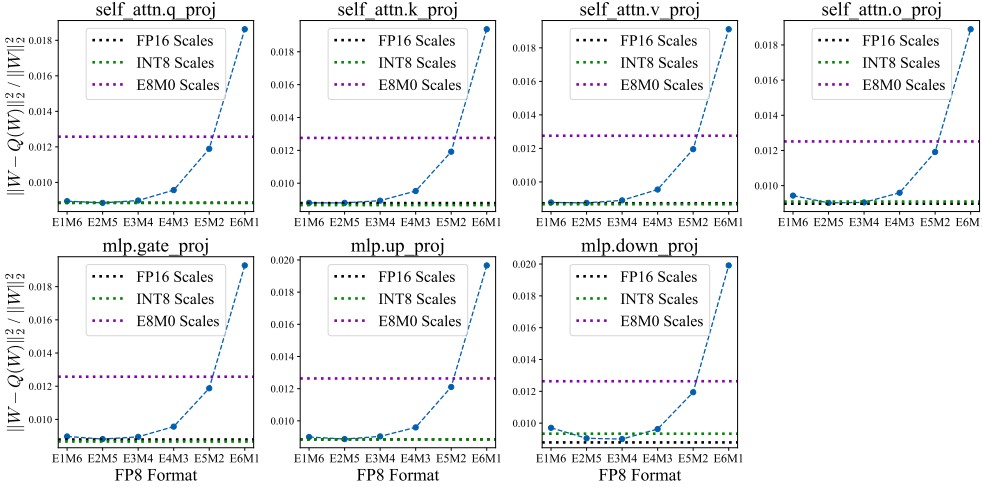

Figure 8: $\text{MSE}^{\text{rel}}$ for the weights of 15th block in the Llama-3.1-8B-Instruct model.

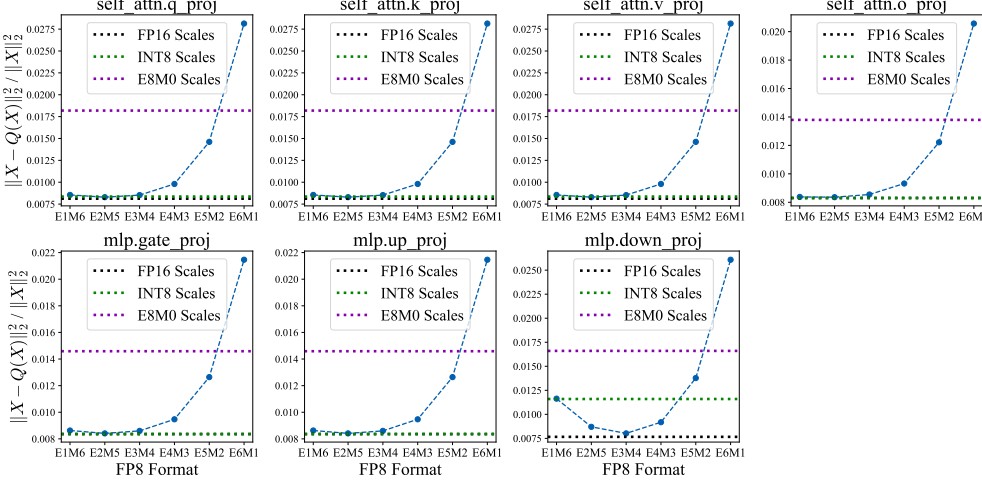

Figure 9: $\text{MSE}^{\text{rel}}$ for the activations of 15th block in the Llama-3.1-8B-Instruct model.

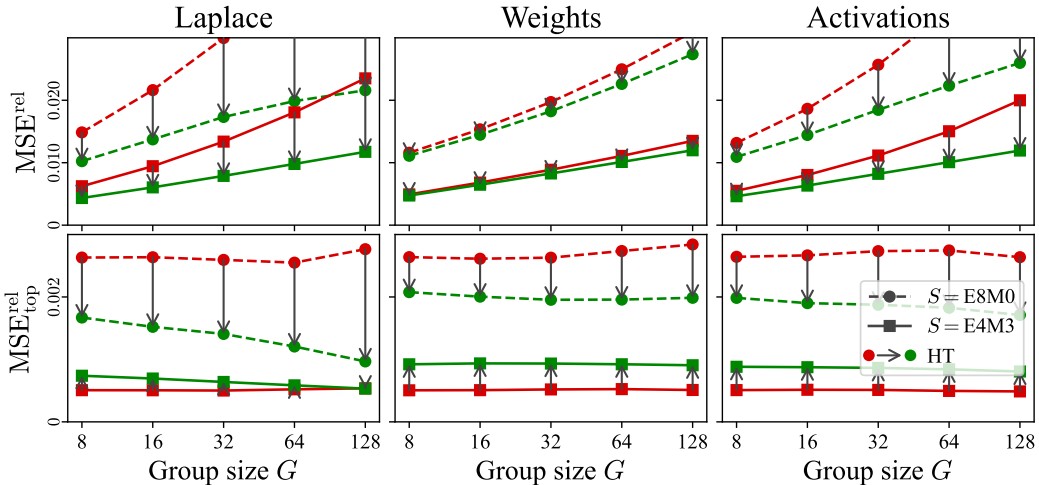

Figure 10: The effect of Hadamard Transform (HT) on MXINT4 (E8M0) and NVINT4 (E4M3) quantization on Laplace distribution samples and Llama-3.1-8B-Instruct weights and activations for various group sizes.

One can observe that the E4M3 and E8M0 scales are not optimal for weight scale quantization. E4M3 and E8M0 increase $\text{MSE}^{\text{rel}}$ by 10%, 40% on average, respectively. At the same time, FP8 options with larger mantissa (E1M6-E3M4) as well as INT8 perform close to FP4 without scale quantization. The pattern for activation pattern is similar except for the case of `down_proj` in feedforward layer, which is known to have a more heavy-tailed distribution with pronounced outliers. We note that the observed behavior generalizes to other models considered in our study.

## D  INT4-BASED MICROSCALING FORMATS

To bridge this analysis gap and inform future hardware designs, we now present a new analysis of *hypothetical* microscaling INT4 formats.

Since INT4 is not a part of the "OCP Microscaling Formats (MX) Specification" [47], we define it ourselves as follows:

1. We define INT4 base data type as a uniform symmetric grid of 15 (to match FP4) elements:
   `[-7, -6, -5, -4, -3, -2, -1, 0, +1, +2, +3, +4, +5, +6, +7]`.

2. We define NVINT4 as a microscaling format with E4M3 shared scales for groups of 16 elements (same as NVFP4) over the INT4 base element data type.

3. We define MXINT4 as a microscaling format with E8M0 shared scales for groups of 32 elements (same as MXFP4) over the INT4 base element data type.

Applying the error analysis from Section 3 to these formats, as demonstrated in Figure 10, reveals that NVINT4 performs close to NVFP4 without the normalizing transforms. Yet, the proposed group Hadamard Transform has positive effect on it (as opposed to negative for NVFP4), making micro-rotated NVINT4 the most accurate of all the analyzed formats. MXINT4, however, performs poorly, and the normalizing transforms yield limited improvement. These findings extrapolate seamlessly to evaluations in Tables 1,3,4 and 5, confirming the usefullness of the Hadamard Transform and the superiority of the micro-rotated NVINT4 format.

## E  OUTLIERS ANALYSIS

**Proof of Lemma 1.** Let $U = \frac{1}{\sqrt{G}}H$ be the normalized Hadamard matrix. $U$ is orthogonal ($U^\top U = I_G$). The error vectors are related by $\varepsilon_x = \widehat{x} - x = U^\top \widehat{y} - U^\top y = U^\top (\widehat{y} - y) = U^\top \varepsilon_y$. Since $U$

is orthogonal, it preserves the Euclidean norm: $\|\varepsilon_x\|_2^2 = \|U^\top \varepsilon_y\|_2^2 = \|\varepsilon_y\|_2^2$. The per-element Mean Squared Error (MSE) is defined as:

$$\text{MSE}(G) = \frac{1}{G}\,\mathbb{E}[\|\varepsilon_x\|_2^2] = \frac{1}{G}\,\mathbb{E}[\|\varepsilon_y\|_2^2].$$

This establishes the second equality.

To prove the first, we rely on the standard assumption in quantization analysis that the quantization error $\varepsilon_y$ is statistically independent of the signal $y$. Since $x$ and $y$ are related by the invertible transformation $x = U^\top y$, $\varepsilon_y$ is also independent of $x$. Consequently, the reconstruction error $\varepsilon_x = U^\top \varepsilon_y$ is also going to be independent of $x$.

The index $I_\star = \arg\max_i |x_i|$ is a function of $x$. Therefore, the error vector $\varepsilon_x$ (and its components) is independent of the random index $I_\star$. Further, since the coordinates of $x$ are i.i.d., we can apply symmetry to obtain that the probability that any coordinate $i$ has the largest magnitude is uniform: $P(I_\star = i) = 1/G$.

We calculate the Top-Element MSE using the Law of Total Expectation:

$$\begin{aligned}
\text{MSE}_{\text{top}}(G) &= \mathbb{E}[(\varepsilon_x)_{I_\star}^2] \\
&= \sum_{i=1}^{G} \mathbb{E}[(\varepsilon_x)_{I_\star}^2 \mid I_\star = i] P(I_\star = i) \\
&= \sum_{i=1}^{G} \mathbb{E}[(\varepsilon_x)_i^2 \mid I_\star = i] \cdot \frac{1}{G}.
\end{aligned}$$

Because $(\varepsilon_x)_i^2$ is independent of the event $\{I_\star = i\}$, the conditional expectation simplifies to $\mathbb{E}[(\varepsilon_x)_i^2 \mid I_\star = i] = \mathbb{E}[(\varepsilon_x)_i^2]$. Substituting yields:

$$\begin{aligned}
\text{MSE}_{\text{top}}(G) &= \frac{1}{G} \sum_{i=1}^{G} \mathbb{E}[(\varepsilon_x)_i^2] \\
&= \frac{1}{G}\mathbb{E}\left[\sum_{i=1}^{G}(\varepsilon_x)_i^2\right] \quad \text{(by linearity of expectation)} \\
&= \frac{1}{G}\mathbb{E}[\|\varepsilon_x\|_2^2] = \text{MSE}(G).
\end{aligned}$$

This completes the proof.

**Lemma 3** (Outliers MAPE). *Let distribution $\mathcal{X}$ be a mix of two distributions: $\mathcal{X}_{base}$ and $\mathcal{X}_{outliers}$ with portions $1 - p$ and $p$ such that:*

*1.* $\min(|\mathcal{X}_{outliers}|) > \max(|\mathcal{X}_{base}|)$,

*2.* $\text{MSE}_{\text{top}}^{\text{rel}}(X \sim \mathcal{X}|X_{I_\star} \sim \mathcal{X}_{outliers}) = \text{MSE}_{\text{top}}^{\text{rel}}(X \sim \mathcal{X}|X_{I_\star} \sim \mathcal{X}_{base})$,

*3.* $p \cdot G \ll \text{MSE}_{\text{top}}^{\text{rel}}(\mathcal{X})$.

*Then the expected outlier relative quadratic error equals $\text{MSE}_{\text{top}}^{\text{rel}}(\mathcal{X})$ up to $O(pG)$:*

$$\mathbb{E}_{X \sim \mathcal{X}}\left[\frac{\sum_{i=1}^{G} \lambda_{X_i \sim \mathcal{X}_{outliers}} \cdot \frac{(X_i - \widehat{X}_i)^2}{X_i^2}}{\sum_{i=1}^{G} \lambda_{X_i \sim \mathcal{X}_{outliers}}}\right] \approx \text{MSE}_{\text{top}}^{\text{rel}}(X \sim \mathcal{X}).$$

*Proof.* We expand the expectation conditioned on $X_{I_\star} \sim \mathcal{X}_{outliers}$:

$$\mathbb{E}_{X \sim \mathcal{X}}\left[\frac{\sum_{i=1}^{G} \lambda_{X_i \sim \mathcal{X}_{outliers}} \cdot \frac{(X_i - \widehat{X}_i)^2}{X_i^2}}{\sum_{i=1}^{G} \lambda_{X_i \sim \mathcal{X}_{outliers}}}\right]$$

$$= \mathbb{E}_{X \sim \mathcal{X} \mid X_{I_\star} \sim \mathcal{X}_{outliers}}\left[\frac{\frac{(X_{I_\star} - \widehat{X}_{I_\star})^2}{X_{I_\star}^2} + \sum_{i \neq I_\star} \lambda_{X_i \sim \mathcal{X}_{outliers}} \cdot \frac{(X_i - \widehat{X}_i)^2}{X_i^2}}{1 + \sum_{i \neq I_\star} \lambda_{X_i \sim \mathcal{X}_{outliers}}}\right]$$

$$= \mathbb{E}_{X \sim \mathcal{X} \mid X_{I_\star} \sim \mathcal{X}_{outliers}}\left[\frac{(X_{I_\star} - \widehat{X}_{I_\star})^2}{X_{I_\star}^2}\right] + O(pG).$$

By Assumption 2 this conditional expectation equals $\mathrm{MSE}_{\mathrm{top}}^{\mathrm{rel}}(\mathcal{X})$, up to $O(pG)$ from Assumption 3. Hence the claim follows. $\square$

**Discussion.** Assumption 1 is satisfied for outliers chosen by absolute value thresholds. Assumption 2 holds for floating-point quantization due to constant relative accuracy (no overflow/underflow), verified in Section 3.2. Assumption 3 holds in practice for LLMs since outliers are typically about 0.1% of elements [12].

**Lemma 4** (Consistency of $\mathrm{MSE}_{\mathrm{top}}^{\mathrm{rel}}$ for smooth distributions). *Let $\mathcal{X}$ be a distribution of values to quantize with a power-of-two translation-invariant quantization function*

$$Q : \forall x \in \mathbb{R}_+, \forall k \in \mathbb{Z} : Q(x \cdot 2^k) = 2^k \cdot Q(x).$$

*Assume:*

1. $\mathrm{supp}\, \mathcal{X} \subset [2^a, 2^b]$ *for integers* $a < b$,

2. $\forall x \in \mathrm{supp}\, \mathcal{X}, \forall y \in [x/\sqrt{2}, x \cdot \sqrt{2}] : |f_{\mathcal{X}}(x) - f_{\mathcal{X}}(y)| \leq \alpha$,

3. $\frac{(x - Q(x))^2}{x^2} \leq \mathrm{MSE}_{\mathrm{max}}^{\mathrm{rel}}$.

*Then*

$$\mathbb{E}_{x \sim \mathcal{X}}\left[\frac{(x - Q(x))^2}{x^2}\right] = \int_1^2 \frac{(x - Q(x))^2}{x^2}\, dx \ + \ O\left((2^b - 2^a)\, \mathrm{MSE}_{\mathrm{max}}^{\mathrm{rel}} \cdot \alpha\right).$$

*Proof.* We decompose the expectation over dyadic intervals:

$$\mathbb{E}_{x \sim \mathcal{X}}\left[\frac{(x - Q(x))^2}{x^2}\right] = \sum_{i=a}^{b-1} \int_{2^i}^{2^{i+1}} \frac{(x - Q(x))^2}{x^2} f_{\mathcal{X}}(x)\, dx.$$

Within each interval, write $f_{\mathcal{X}}(x) = f_{\mathcal{X}}(2^i) + (f_{\mathcal{X}}(x) - f_{\mathcal{X}}(2^i))$. The first term yields

$$\int_1^2 \frac{(x - Q(x))^2}{x^2}\, dx \cdot \sum_{i=a}^{b-1} 2^i f_{\mathcal{X}}(2^i).$$

The second term is bounded using Assumption 2 and 3, giving

$$\sum_{i=a}^{b-1} \int_{2^i}^{2^{i+1}} \mathrm{MSE}_{\mathrm{max}}^{\mathrm{rel}} \cdot O(\alpha)\, dx = (2^b - 2^a) \cdot \mathrm{MSE}_{\mathrm{max}}^{\mathrm{rel}} \cdot O(\alpha).$$

Finally, the normalization error in the discrete approximation of $\int f_{\mathcal{X}}$ contributes an additional $O(\alpha)$ factor. Combining terms gives the stated result. $\square$

**Discussion.** Assumptions 1 and 3 hold for `absmax` $X_{I_\star}$ quantization since floating-point values are bounded with bounded relative error. Assumption 2 is supported empirically (Figure 4), where scale distributions are observed to be smooth.

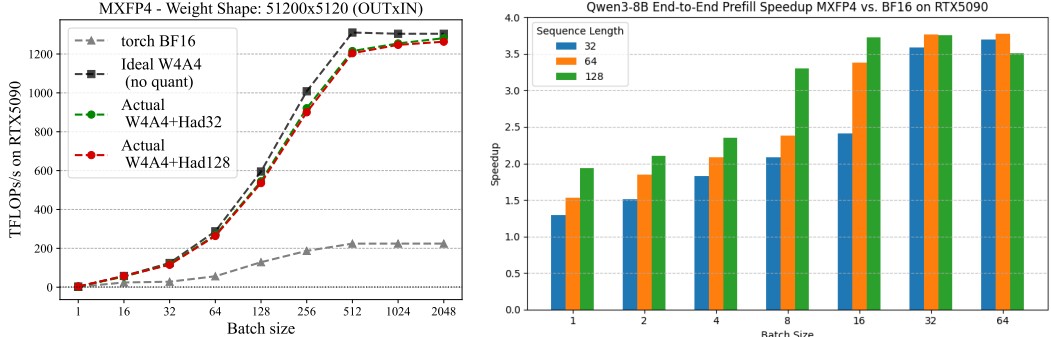

Figure 11: Illustration of QuTLASS performance for weights and activations on MXFP4 while increasing batch size, for a single linear LLM layer, showing the low-overhead of the quantization-related ops, and end-to-end using the Transformers library.

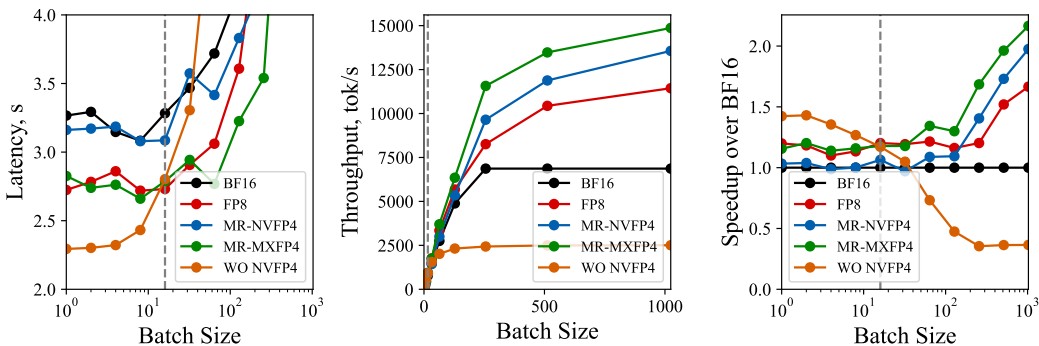

Figure 12: End-to-end speedups for Llama-3.3-70B-Instruct. The gray vertical line roughly separates small-batch inference (which covers single-user text generation) from large-batch inference (which includes prefill).

## F  QUTLASS RESULTS ON GEFORCE GPUS

Figure 11 illustrates additional QuTLASS performance results on an NVIDIA RTX5090 GPU. The figure on the left shows throughput for a single layer extracted from a MXPF4 quantized Qwen3-32B model, while the figure on the right shows the end-to-end speedups on Transformers running Qwen3-8B with MXFP4 quantization compared to the BF16 baseline implementation on a single RTX5090 GPU.

## G  QUTLASS SPEEDUPS IN VARIOUS INFERENCE REGIMES

Figure 12 demonstrates end-to-end inference speedups of FP8 and various FP4 formats relative to BF16 for small and large batch sizes for Llama-3.3-70B-Instruct running on a single NVIDIA B200 GPU in vLLM.

For large batch workloads, it is easy to see how our Micro-Rotated (MR) kernels outperform both BF16 and FP8, with MR-MXFP4 providing the highest throughput at around 15,000 tok/sec: a 2.2x increase over BF16 and a 1.3x increase over FP8.

These kernels, however, were not optimized for small batch workloads, where they show little to no improvement over BF16 and FP8. In that regime, which is characterized by memory-bound inference, it is preferable to use weight-only quantization, as activation quantization brings no benefits to inference speed. We present accuracy measurements of weight-only quantized models in Appendix A. In Figure 12, we present latency and throughput measurements for a weight-only (WO)

micro-rotated FP4 quantization scheme, which is already supported in vLLM. One can see that it shows approximately 20% lower latency than FP8 for small batch size inference.

# H    MXFP SCALE FITTING

The principal cause of the poor performance of MXFP quantization is the large-scale quantization error. On the one hand, the E8M0 format allows representing extremely small values, such as $2^{-127}$, and extremely large values, such as $2^{128}$. On the other, as shown in Figure 4, the actual range of weights and activations is much more narrow, making the E8M0 grid too coarse. To address this mismatch, we propose a simple modification: fit the quantization grid to the data range.

The original MXFP quantization grid, with 4/3 re-scaling, quantizes scales as follows:

$$s_{\text{E8M0}} = (4/3) \cdot 2^{\text{clamp}(\text{round}(\log_2 s), -128, 127)}. \tag{1}$$

One can estimate the smallest $s_{\min}$ and the largest $s_{\max}$ values in given tensors, such that the smallest value is mapped to -128, and the largest to 127:

$$s_{\text{E8M0}} = 2^{(\log_2 s_{\max} - \log_2 s_{\min})\text{clamp}\left(\text{round}\left(255 \cdot \frac{\log_2 s - \log_2 s_{\min}}{\log_2 s_{\max} - \log_2 s_{\min}}\right), 0, 255\right) + \log_2 s_{\min}}. \tag{2}$$

The modification reduces the scale-quantization error, by rescaling the exponent. In effect, we replace the exponent by a value $2^{\alpha}, \alpha \in (0, 1)$, followed by rescaling:

$$s = 2^{\alpha q + \beta} \tag{3}$$

We provide the comparision between the original MXFP and modified version, called MXFP4$^\dagger$, in Table 11. In almost every setting, MXFP4$^\dagger$ yields a substantial performance boost relative to vanilla MXFP4 and performs close to NVFP4. One should note that MXFP4$^\dagger$ requires 4.25 bit per parameter compared to 4.5 for NVFP.

| Method | Format | Llama3 (8B) | Qwen3 (8B) |
|---|---|---|---|
| RTN | MXFP4 | 87.8 | 93.7 |
| | MXFP4$^\dagger$ | 94.3 (+6.5) | 96.3 (+2.6) |
| | NVFP4 | 94.7 | 98.9 |
| RTN + HT | MXFP4 | 89.2 | 93.6 |
| | MXFP4$^\dagger$ | 93.9 (+4.7) | 96.3 (+2.7) |
| | NVFP4 | 93.8 | 96.0 |
| GPTQ | MXFP4 | 89.5 | 94.1 |
| | MXFP4$^\dagger$ | 95.2 (+5.7) | 92.3 (-1.8) |
| | NVFP4 | 95.7 | 98.1 |
| MR-GPTQ | MXFP4 | 93.6 | 95.2 |
| | MXFP4$^\dagger$ | 94.9 (+1.3) | 98.5 (+3.3) |
| | NVFP4 | 95.8 | 97.4 |

Table 11: Per-model recoveries with vanilla MXFP4 format and the proposed modification denoted by MXFP4$^\dagger$.

Efficient arithmetic with MXFP scales requires that the weight and activation tensors in a layer share the same exponent. Consequently, the product of the activation scale $s_A$ and the weight scale $s_W$ can be expressed as:

$$s_A s_W = 2^{\alpha_A q_A + \beta_A} 2^{\alpha_W q_W + \beta_W} = 2^{\alpha_A q_A + \beta_A + \alpha_B q_B + \beta_B}. \tag{4}$$

In the above expression, $\alpha_A$ and $\alpha_B$ have to the same for one to express scale multiplication in terms of power addition.

# I    COMPARISON BETWEEN VARIANTS ON THE PLATINUM BENCHMARK

Next, we perform a detailed analysis between different versions of the GPTQ algorithm, on the two formats. As visible in Table 1, the standard evaluation harness struggles to distinguish variants, likely

because of high noise in some of the evaluations. To address this, we examine the differences between GPTQ variants on the less noisy PlatinumBench benchmark [53], which includes carefully curated tasks and questions. These experiments are performed with "real" kernels, via our vLLM integration. Figure 13 include full results across tasks, for the following variants:

- **Transform matrices**: Hadamard with different block sizes, denoted by e.g. **Had128**.
- **Scale optimization**: Our approach (**MSE**) or the default (**MinMax**).
- **Quantization ordering**: Static Activation Ordering (**ActOrder**) or arbitrary/initial (**Default**).

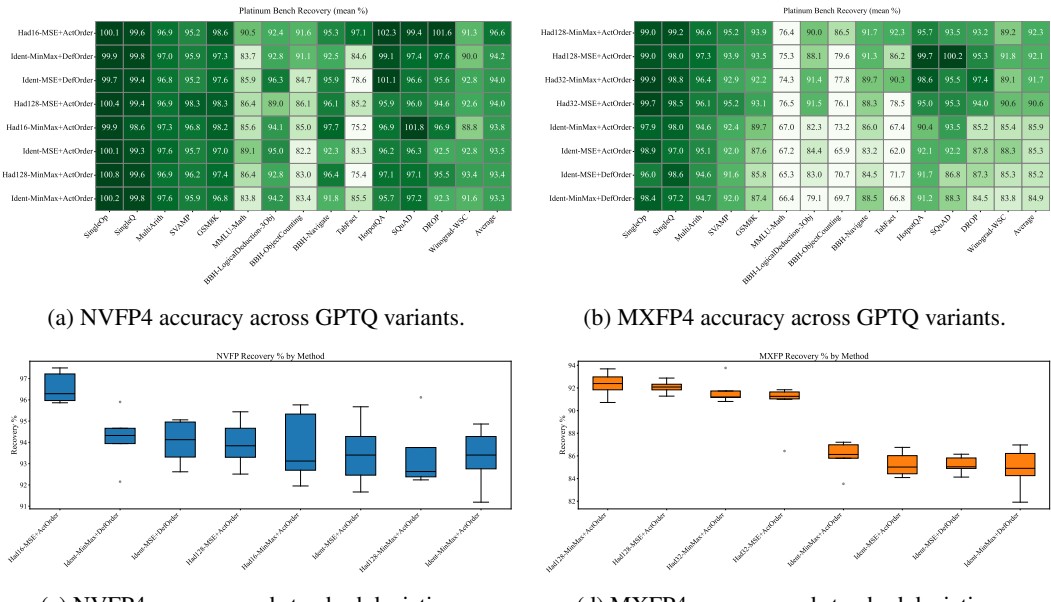

(a) NVFP4 accuracy across GPTQ variants.

(b) MXFP4 accuracy across GPTQ variants.

(c) NVFP4 averages and standard deviations.

(d) MXFP4 averages and standard deviations.

Figure 13: Comparison of NVFP4 and MXFP4 quantization methods on Platinum benchmark tasks. Top row shows recovery results across different GPTQ/MR-GPTQ component combinations. Bottom row shows average recovery scores and standard deviations for each method.

**Discussion.** We observe the following:

- The results suggest that Hadamard rotations provide a statistically-significant advantage to our MR-GPTQ variant, with group-aligned Hadmard rotations (Had16), MSE and ActOrder, in the NVFP4 case as well. All other variants appear to be within variance of eachother on this benchmark, for NVFP4.
- We observe a large gap ( > 4 points on average) between the top NVFP4 recovery (96.6%) and the top MXFP4 recovery (92.3%).
- Finally, the MXFP4 results show a very large recovery gap between the variants with rotations and the variants without. Moreover, for MXFP4, larger Hadamard rotations (128 vs 32) appear to clearly help, whereas, for NVFP4, matching the rotation size to the group size appears ideal.

## J  STANDARD DEVIATION

We estimate the variance of evaluation scores by performing multiple quantization runs on Llama-3.1-8B-Instruct, varying the seeds for GPTQ calibration set sampling, as well as the strategies for scale selection and quantization ordering. These results were generated using our vLLM integration with QuTLASS kernels. Figure 14 displays the scores as bar plots, while Table 12 lists the average recovery scores and their standard deviations.

Additionally, we report the average recovery scores and their standard deviations for the Platinum benchmark suite [53] in Figure 17 and Table 12. We also report per-task recovery (%) in Figure 15.

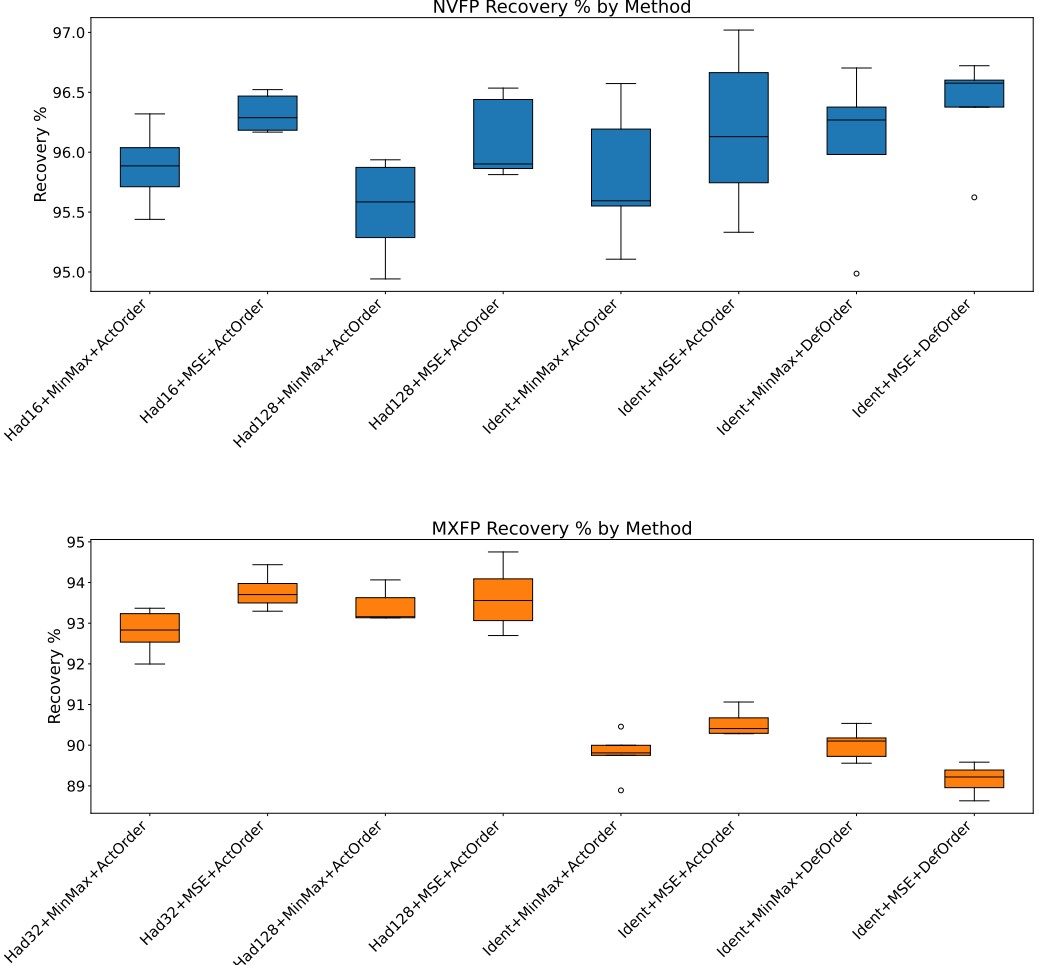

Figure 14: Accuracy results for NVFP4 and MXFP4 across different combinations of MR-GPTQ components, averaged over five random seeds using vLLM kernels on the benchmark suite.

Figure 15: Accuracy results for MXFP4 across different MR-GPTQ component combinations on the Platinum benchmark tasks.

Platinum Bench Recovery (mean %)

| | SingleOp | SingleQ | MultiArith | SVAMP | GSM8K | MMLU-Math | BBH-LogicalDeduction-3Obj | BBH-ObjectCounting | BBH-Navigate | TabFact | HotpotQA | SQuAD | DROP | Winograd-WSC | Average |
|---|---|---|---|---|---|---|---|---|---|---|---|---|---|---|---|
| Had128-MinMax+ActOrder | 99.0 | 99.2 | 96.6 | 95.2 | 93.9 | 76.4 | 90.0 | 86.5 | 91.7 | 92.3 | 95.7 | 93.5 | 93.2 | 89.2 | 92.3 |
| Had128-MSE+ActOrder | 99.0 | 98.0 | 97.3 | 93.9 | 93.5 | 75.3 | 88.1 | 79.6 | 91.3 | 86.2 | 99.7 | 100.2 | 95.3 | 91.8 | 92.1 |
| Had32-MinMax+ActOrder | 99.9 | 98.8 | 96.4 | 92.9 | 92.2 | 74.3 | 91.4 | 77.8 | 89.7 | 90.3 | 98.6 | 95.5 | 97.4 | 89.1 | 91.7 |
| Had32-MSE+ActOrder | 99.7 | 98.5 | 96.1 | 95.2 | 93.1 | 76.5 | 91.5 | 76.1 | 88.3 | 78.5 | 95.0 | 95.3 | 94.0 | 90.6 | 90.6 |
| Ident-MinMax+ActOrder | 97.9 | 98.0 | 94.6 | 92.4 | 89.7 | 67.0 | 82.3 | 73.2 | 86.0 | 67.4 | 90.4 | 93.5 | 85.2 | 85.4 | 85.9 |
| Ident-MSE+ActOrder | 98.9 | 97.0 | 95.1 | 92.0 | 87.6 | 67.2 | 84.4 | 65.9 | 83.2 | 62.0 | 92.1 | 92.2 | 87.8 | 88.3 | 85.3 |
| Ident-MSE+DefOrder | 96.0 | 98.6 | 94.6 | 91.6 | 85.8 | 65.3 | 83.0 | 70.7 | 84.5 | 71.7 | 91.7 | 86.8 | 87.3 | 85.3 | 85.2 |
| Ident-MinMax+DefOrder | 98.4 | 97.2 | 94.7 | 92.0 | 87.4 | 66.4 | 79.1 | 69.7 | 88.5 | 66.8 | 91.2 | 88.3 | 84.5 | 83.8 | 84.9 |

Figure 16: Accuracy results for NVFP4 across different MR-GPTQ component combinations on the Platinum benchmark tasks.

Platinum Bench Recovery (mean %)

| | SingleOp | SingleQ | MultiArith | SVAMP | GSM8K | MMLU-Math | BBH-LogicalDeduction-3Obj | BBH-ObjectCounting | BBH-Navigate | TabFact | HotpotQA | SQuAD | DROP | Winograd-WSC | Average |
|---|---|---|---|---|---|---|---|---|---|---|---|---|---|---|---|
| Had16-MSE+ActOrder | 100.1 | 99.6 | 96.9 | 95.2 | 98.6 | 90.5 | 92.4 | 91.6 | 95.3 | 97.1 | 102.3 | 99.4 | 101.6 | 91.3 | 96.6 |
| Ident-MinMax+DefOrder | 99.9 | 99.8 | 97.0 | 95.9 | 97.3 | 83.7 | 92.8 | 91.1 | 92.5 | 84.6 | 99.1 | 97.4 | 97.6 | 90.0 | 94.2 |
| Ident-MSE+DefOrder | 99.7 | 99.4 | 96.8 | 95.2 | 97.6 | 85.9 | 96.3 | 84.7 | 95.9 | 78.6 | 101.1 | 96.6 | 95.6 | 92.8 | 94.0 |
| Had128-MSE+ActOrder | 100.4 | 99.4 | 96.9 | 98.3 | 98.3 | 86.4 | 89.0 | 86.1 | 96.1 | 85.2 | 95.9 | 96.0 | 94.6 | 92.6 | 94.0 |
| Had16-MinMax+ActOrder | 99.9 | 98.6 | 97.3 | 96.8 | 98.2 | 85.6 | 94.1 | 85.0 | 97.7 | 75.2 | 96.9 | 101.8 | 96.9 | 88.8 | 93.8 |
| Ident-MSE+ActOrder | 100.1 | 99.3 | 97.6 | 95.7 | 97.0 | 89.1 | 95.0 | 82.2 | 92.3 | 83.3 | 96.2 | 96.3 | 92.5 | 92.8 | 93.5 |
| Had128-MinMax+ActOrder | 100.8 | 99.6 | 96.9 | 96.2 | 97.4 | 86.4 | 92.8 | 83.0 | 96.4 | 75.4 | 97.1 | 97.1 | 95.5 | 93.4 | 93.4 |
| Ident-MinMax+ActOrder | 100.2 | 99.8 | 97.6 | 95.9 | 96.8 | 83.8 | 94.2 | 83.4 | 91.8 | 85.5 | 95.7 | 97.2 | 92.3 | 91.6 | 93.3 |

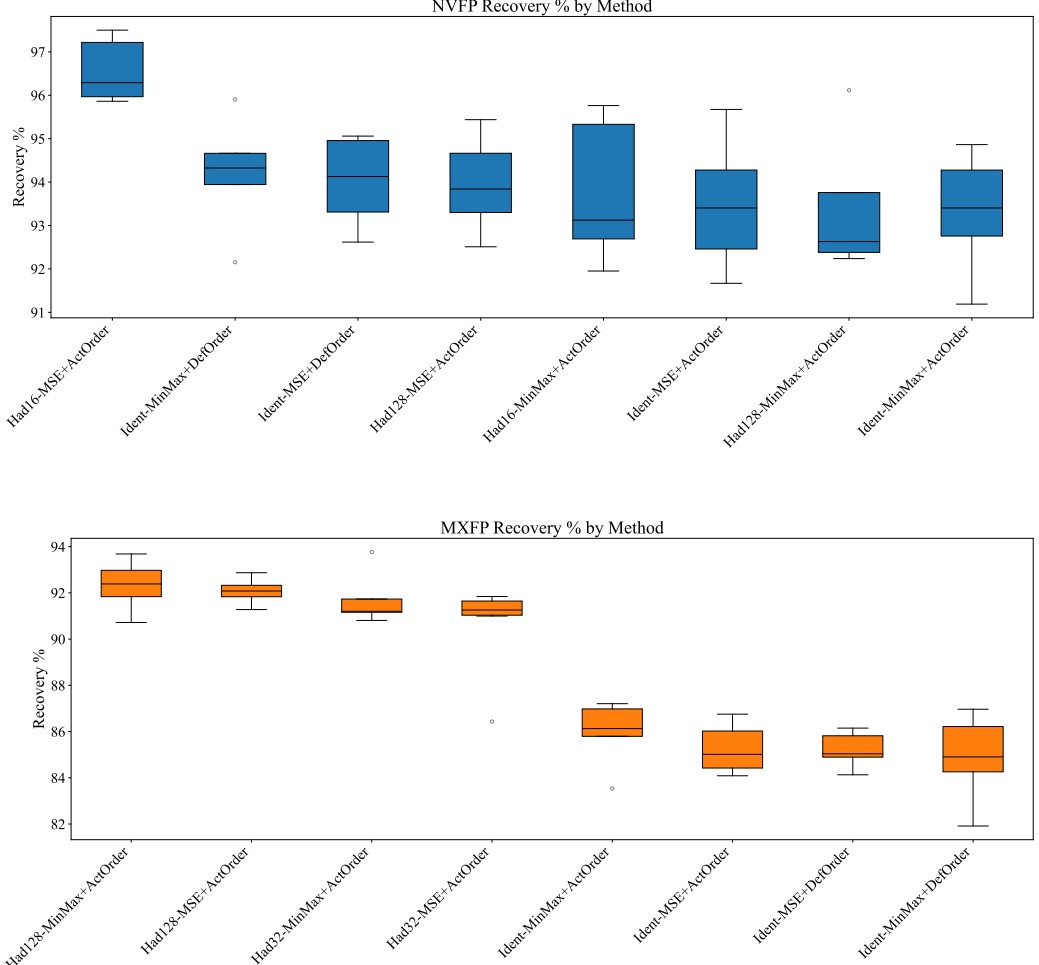

Figure 17: Average recovery scores and standard deviations for NVFP and MXFP methods on the Platinum benchmarks.

| Format | Method | Standard Bench % | STD | Platinum Bench % | STD |
|---|---|---|---|---|---|
| NVFP | Had16+MinMax+ActOrder | 95.88 | 0.332 | 93.77 | 1.680 |
| | Had16+MSE+ActOrder | 96.33 | 0.163 | 96.57 | 0.746 |
| | Had128+MinMax+ActOrder | 95.52 | 0.416 | 93.42 | 1.618 |
| | Had128+MSE+ActOrder | 96.11 | 0.347 | 93.95 | 1.143 |
| | Ident+MinMax+ActOrder | 95.84 | 0.487 | 93.27 | 1.263 |
| | Ident+MSE+ActOrder | 96.18 | 0.589 | 93.51 | 1.304 |
| | Ident+MinMax+DefOrder | 96.06 | 0.655 | 94.20 | 1.358 |
| | Ident+MSE+DefOrder | 96.38 | 0.441 | 94.01 | 1.053 |
| MXFP | Had32+MinMax+ActOrder | 92.79 | 0.554 | 91.73 | 1.183 |
| | Had32+MSE+ActOrder | 93.78 | 0.445 | 90.51 | 2.294 |
| | Had128+MinMax+ActOrder | 93.42 | 0.416 | 92.32 | 1.128 |
| | Had128+MSE+ActOrder | 93.63 | 0.817 | 91.86 | 0.743 |
| | Ident+MinMax+ActOrder | 89.78 | 0.570 | 85.93 | 1.459 |
| | Ident+MSE+ActOrder | 90.54 | 0.330 | 85.26 | 1.112 |
| | Ident+MinMax+DefOrder | 89.16 | 0.372 | 84.85 | 1.95 |
| | Ident+MSE+DefOrder | 90.02 | 0.387 | 85.21 | 0.798 |

Table 12: Average Recovery scores and standard deviations for NVFP and MXFP methods across the Standard benchmark and Platinum benchmark.

## K    THE EFFECT OF DIFFERENT LINEAR TRANSFORMS

In this section we ablate various choices of transforms adopted for outlier mitigation of outliers. Specifically, we consider the following options:

- Identity transform.
- Discrete Cosine Transform (DCT).
- Hadamard rotation [50; 4; 11].
- Grouped Sequency-arranged Rotation (GSR) [8].

We sweep over different options of transform sizes ($\{16, 32, 64, 128, 256\}$) both for NVFP and MXFP formats. The average score on 5 tasks from LM Evaluation Harness (piqa, winogrande, hellaswag, arc-easy, arc-challenge) is reported.

From these results, one can observe that rotations yield small improvement relative to identity transform for MXFP format and minor degradation for NVFP with RTN quantization. Different transform sizes perform more or less the same.

| Transformation | Transformation Size | Weight Quant | PIQA | winogrande | hellaswag | arc-easy | arc-challenge | Avg |
|---|---|---|---|---|---|---|---|---|
| FP16 | - | - | 0.8074 | 0.7301 | 0.792 | 0.7769 | 0.5307 | 0.7274 |
| - | - | RTN | 0.802 | 0.7261 | 0.7731 | 0.7466 | 0.4923 | 0.708 |
|  | - | GPTQ | 0.7933 | 0.7214 | 0.7698 | 0.7664 | 0.5111 | 0.7124 |
| DCT | 16 | RTN | 0.79 | 0.6859 | 0.7583 | 0.742 | 0.4889 | 0.693 |
|  |  | GPTQ | 0.7824 | 0.7111 | 0.766 | 0.7559 | 0.4991 | 0.7029 |
|  | 32 | RTN | 0.7786 | 0.7119 | 0.7572 | 0.7353 | 0.4693 | 0.6905 |
|  |  | GPTQ | 0.7813 | 0.7135 | 0.7718 | 0.7117 | 0.4829 | 0.6922 |
|  | 64 | RTN | 0.7862 | 0.7024 | 0.7695 | 0.7306 | 0.4599 | 0.6897 |
|  |  | GPTQ | 0.7878 | 0.7198 | 0.7673 | 0.7765 | 0.5068 | 0.7116 |
|  | 128 | RTN | 0.7737 | 0.7206 | 0.7676 | 0.7466 | 0.4701 | 0.6957 |
|  |  | GPTQ | 0.7873 | 0.708 | 0.7715 | 0.7399 | 0.494 | 0.7001 |
|  | 256 | RTN | 0.7916 | 0.7135 | 0.7698 | 0.7563 | 0.4983 | 0.7059 |
|  |  | GPTQ | 0.7911 | 0.7017 | 0.7692 | 0.7694 | 0.506 | 0.7074 |
| DST | 16 | RTN | 0.7824 | 0.7143 | 0.7575 | 0.7256 | 0.4804 | 0.692 |
|  |  | GPTQ | 0.7878 | 0.7198 | 0.7628 | 0.7395 | 0.4855 | 0.6991 |
|  | 32 | RTN | 0.7856 | 0.7198 | 0.7399 | 0.7395 | 0.4667 | 0.6903 |
|  |  | GPTQ | 0.7889 | 0.7096 | 0.7633 | 0.7731 | 0.5026 | 0.7075 |
|  | 64 | RTN | 0.7911 | 0.7253 | 0.7536 | 0.7635 | 0.4804 | 0.7028 |
|  |  | GPTQ | 0.7911 | 0.7088 | 0.7638 | 0.7614 | 0.5 | 0.705 |
|  | 128 | RTN | 0.7856 | 0.7024 | 0.7625 | 0.7677 | 0.4881 | 0.7013 |
|  |  | GPTQ | 0.7824 | 0.7064 | 0.7637 | 0.7778 | 0.5009 | 0.7062 |
|  | 256 | RTN | 0.7867 | 0.6993 | 0.7579 | 0.737 | 0.4804 | 0.6923 |
|  |  | GPTQ | 0.7856 | 0.7048 | 0.7674 | 0.7462 | 0.4812 | 0.6971 |
| Hadamard | 16 | RTN | 0.7927 | 0.7096 | 0.7674 | 0.7471 | 0.465 | 0.6963 |
|  |  | GPTQ | 0.7873 | 0.7096 | 0.7697 | 0.758 | 0.5034 | 0.7056 |
|  | 32 | RTN | 0.784 | 0.719 | 0.7639 | 0.7534 | 0.4881 | 0.7017 |
|  |  | GPTQ | 0.7965 | 0.7348 | 0.7668 | 0.7538 | 0.506 | 0.7116 |
|  | 64 | RTN | 0.7818 | 0.7032 | 0.763 | 0.7395 | 0.4863 | 0.6948 |
|  |  | GPTQ | 0.7856 | 0.7151 | 0.7657 | 0.7614 | 0.5017 | 0.7059 |
|  | 128 | RTN | 0.7884 | 0.7206 | 0.766 | 0.7551 | 0.506 | 0.7072 |
|  |  | GPTQ | 0.7938 | 0.7111 | 0.7729 | 0.7681 | 0.5273 | 0.7146 |
|  | 256 | RTN | 0.7878 | 0.6969 | 0.7643 | 0.7681 | 0.4983 | 0.7031 |
|  |  | GPTQ | 0.79 | 0.7253 | 0.7738 | 0.7673 | 0.4949 | 0.7102 |
| GSR | 16 | RTN | 0.7933 | 0.7056 | 0.7694 | 0.7513 | 0.4744 | 0.6988 |
|  |  | GPTQ | 0.7998 | 0.6985 | 0.7683 | 0.7635 | 0.4863 | 0.7033 |
|  | 32 | RTN | 0.7873 | 0.6985 | 0.762 | 0.7702 | 0.494 | 0.7024 |
|  |  | GPTQ | 0.79 | 0.7214 | 0.77 | 0.7593 | 0.5 | 0.7081 |
|  | 64 | RTN | 0.7911 | 0.7151 | 0.7627 | 0.7588 | 0.4821 | 0.702 |
|  |  | GPTQ | 0.796 | 0.7222 | 0.7717 | 0.7622 | 0.4949 | 0.7094 |
|  | 128 | RTN | 0.7878 | 0.7174 | 0.7656 | 0.7546 | 0.4898 | 0.703 |
|  |  | GPTQ | 0.7894 | 0.7143 | 0.7721 | 0.7668 | 0.506 | 0.7097 |
|  | 256 | RTN | 0.7797 | 0.6906 | 0.7626 | 0.7454 | 0.4735 | 0.6904 |
|  |  | GPTQ | 0.8014 | 0.7293 | 0.7756 | 0.763 | 0.4991 | 0.7137 |

Table 13: Performance of Llama-3-8B with different transformations with NVFP4 format.

| Transformation | Transformation Size | Weight Quant | PIQA | winogrande | hellaswag | arc-easy | arc-challenge | Avg |
|---|---|---|---|---|---|---|---|---|
| FP16 | - | - | 0.8074 | 0.7301 | 0.792 | 0.7769 | 0.5307 | 0.7274 |
| - | - | RTN | 0.7704 | 0.6875 | 0.7481 | 0.7121 | 0.471 | 0.6778 |
| | - | GPTQ | 0.7699 | 0.693 | 0.753 | 0.7327 | 0.4718 | 0.6841 |
| DCT | 16 | RTN | 0.7628 | 0.7072 | 0.7447 | 0.7205 | 0.4582 | 0.6787 |
| | | GPTQ | 0.7753 | 0.7009 | 0.7534 | 0.7365 | 0.4846 | 0.6902 |
| | 32 | RTN | 0.7699 | 0.6914 | 0.7405 | 0.6987 | 0.4437 | 0.6688 |
| | | GPTQ | 0.7508 | 0.6969 | 0.7465 | 0.7281 | 0.4531 | 0.6751 |
| | 64 | RTN | 0.7693 | 0.7127 | 0.7454 | 0.7079 | 0.4556 | 0.6782 |
| | | GPTQ | 0.7889 | 0.7111 | 0.7524 | 0.7529 | 0.465 | 0.6941 |
| | 128 | RTN | 0.7541 | 0.6851 | 0.7398 | 0.6616 | 0.4036 | 0.6488 |
| | | GPTQ | 0.7731 | 0.7088 | 0.7455 | 0.7462 | 0.4804 | 0.6908 |
| | 256 | RTN | 0.7791 | 0.6953 | 0.7392 | 0.6987 | 0.4411 | 0.6707 |
| | | GPTQ | 0.7894 | 0.6946 | 0.7541 | 0.7504 | 0.4744 | 0.6926 |
| DST | 16 | RTN | 0.7731 | 0.6906 | 0.7493 | 0.7391 | 0.4522 | 0.6809 |
| | | GPTQ | 0.7835 | 0.6898 | 0.7593 | 0.7399 | 0.4787 | 0.6902 |
| | 32 | RTN | 0.7639 | 0.6906 | 0.7441 | 0.7332 | 0.4582 | 0.678 |
| | | GPTQ | 0.7802 | 0.6985 | 0.7563 | 0.7483 | 0.4753 | 0.6917 |
| | 64 | RTN | 0.7704 | 0.689 | 0.7402 | 0.7054 | 0.4599 | 0.673 |
| | | GPTQ | 0.7769 | 0.6875 | 0.7599 | 0.7189 | 0.4693 | 0.6825 |
| | 128 | RTN | 0.7612 | 0.6772 | 0.7491 | 0.7003 | 0.4497 | 0.6675 |
| | | GPTQ | 0.7693 | 0.6914 | 0.7567 | 0.7462 | 0.d923 | 0.6912 |
| | 256 | RTN | 0.7731 | 0.6906 | 0.7493 | 0.7391 | 0.4522 | 0.6809 |
| | | GPTQ | 0.778 | 0.7064 | 0.7544 | 0.7412 | 0.4923 | 0.6945 |
| Hadamard | 16 | RTN | 0.7737 | 0.6906 | 0.7499 | 0.6995 | 0.4616 | 0.675 |
| | | GPTQ | 0.7867 | 0.7206 | 0.7623 | 0.7218 | 0.4701 | 0.6923 |
| | 32 | RTN | 0.7715 | 0.6946 | 0.7518 | 0.7466 | 0.5034 | 0.6936 |
| | | GPTQ | 0.7807 | 0.7032 | 0.763 | 0.7471 | 0.4778 | 0.6944 |
| | 64 | RTN | 0.7862 | 0.7088 | 0.7511 | 0.7315 | 0.4667 | 0.6889 |
| | | GPTQ | 0.796 | 0.6993 | 0.7625 | 0.7635 | 0.4923 | 0.7027 |
| | 128 | RTN | 0.7807 | 0.7064 | 0.7529 | 0.7306 | 0.4548 | 0.6851 |
| | | GPTQ | 0.7807 | 0.6946 | 0.7646 | 0.7538 | 0.4915 | 0.697 |
| | 256 | RTN | 0.778 | 0.7024 | 0.7491 | 0.7104 | 0.4625 | 0.6805 |
| | | GPTQ | 0.7818 | 0.7032 | 0.7624 | 0.7576 | 0.4795 | 0.6969 |
| GSR | 16 | RTN | 0.7813 | 0.6977 | 0.7522 | 0.6982 | 0.4684 | 0.6796 |
| | | GPTQ | 0.7845 | 0.7048 | 0.7682 | 0.7546 | 0.4735 | 0.6971 |
| | 32 | RTN | 0.7748 | 0.693 | 0.7514 | 0.742 | 0.4991 | 0.6921 |
| | | GPTQ | 0.7856 | 0.7111 | 0.7631 | 0.7517 | 0.5026 | 0.7028 |
| | 64 | RTN | 0.7889 | 0.7072 | 0.7464 | 0.7226 | 0.4514 | 0.6833 |
| | | GPTQ | 0.7949 | 0.7009 | 0.7613 | 0.7412 | 0.4829 | 0.6962 |
| | 128 | RTN | 0.7753 | 0.7001 | 0.7538 | 0.7226 | 0.4659 | 0.6835 |
| | | GPTQ | 0.7813 | 0.7056 | 0.7595 | 0.7395 | 0.4846 | 0.6941 |
| | 256 | RTN | 0.7753 | 0.6819 | 0.7494 | 0.7197 | 0.4642 | 0.6781 |
| | | GPTQ | 0.778 | 0.7151 | 0.7542 | 0.7445 | 0.4923 | 0.6968 |

Table 14: Performance of Llama-3-8B with different transformations with MXFP4 format.

