# OpenReview forum: "Bridging the Gap Between Promise and Performance for Microscaling FP4 Quantization"
_ICLR.cc/2026/Conference — ICLR 2026 Poster_

### Official Review · Reviewer_WXdo · 2025-10-23

**Soundness:** 3
**Presentation:** 3
**Contribution:** 3
**Rating:** 8
**Confidence:** 4

**Summary:**

This paper provides the first comprehensive study of FP4 microscaling floating-point formats—MXFP4 and NVFP4—used in recent NVIDIA and AMD GPUs for LLM quantization. The authors analyze why these formats underperform despite their hardware support: (1) NVFP4’s small group size limits outlier mitigation, and (2) MXFP4’s power-of-two scale quantization causes high quantization error. To address this, they introduce Micro-Rotated GPTQ (MR-GPTQ), a GPTQ variant tailored to FP4, using block-wise Hadamard rotations, MSE-optimized scales, and static activation reordering. They implement this through QuTLASS, a GPU kernel suite supporting fused online rotations with negligible overhead. Experiments on Llama and Qwen models show that MR-GPTQ significantly improves MXFP4 accuracy (closing most of the gap to NVFP4) and achieves up to 3.6×–6× layerwise and 2.2×–4× end-to-end speedups on Blackwell GPUs. The paper concludes that FP4 is not inherently superior to INT4, but format-specific optimization can unlock its full potential.

**Strengths:**

1. The paper tackles an important and emerging problem—quantization under new FP4 microscaling formats (MXFP4 and NVFP4)—that directly relates to current-generation GPU hardware (NVIDIA Blackwell, AMD CDNA4). This makes the work highly relevant for both research and deployment.
2. The paper implements QuTLASS, optimized GPU kernels that fuse quantization, rotation, and matrix multiplication for FP4 inference. Integrated with the Blackwell tensor core pipeline, these kernels add almost no overhead, effectively bridging algorithm design and hardware deployment.
3. The authors present a clear analytical treatment of quantization error for FP4, explaining the contrasting effects of rotations under Laplace vs. Normal distributions, and showing why existing PTQ methods (e.g., GPTQ, SmoothQuant) underperform on FP4 formats.

**Weaknesses:**

1. Despite the proposed improvements, FP4 quantization—especially with MXFP4—still suffers from noticeable accuracy degradation. Even with MR-GPTQ, the recovery remains 3–5 % below FP16 for many models, suggesting that FP4 formats are not yet competitive with INT4 or FP8 in accuracy.
2. The methodological novelty is moderate, as MR-GPTQ mainly adapts the existing GPTQ framework to FP4 formats using known ideas such as block-wise rotations and scale optimization. However, the work shows strong engineering and system-level originality—it is the first to deeply study NVFP4 and MXFP4, propose an FP4-specific GPTQ variant, and implement efficient GPU kernels with near-zero overhead. Overall, the contribution is not conceptually new but is practically impactful and timely.

**Questions:**

The current speed evaluation focuses mainly on end-to-end throughput. Is there any latency measurements for the prefill and decode stages, respectively?

---

> ### Author Response · Authors · 2025-11-17
>
> Thank you for your review!
>
> We discuss the aforementioned weaknesses below:
>
> > FP4 formats are not yet competitive with INT4 or FP8 in accuracy
>
> For 4-bit formats, Table 1 shows that NVFP4 does outperform INT4 quantization and Figure 4 shows that MXFP4 augmented with QAT also outperforms INT4 baselines from Table 1. MXFP4 does show worse accuracy than NVFP4, but, in our assessment, it certainly is competitive with other 4-bit formats.
>
> For FP8, it’s important to remember that it can’t be directly compared with 4-bit formats because they offer substantially different FLOPs and model throughputs on modern accelerators. It is also worth noting that the performance gap decreases as the model size grows. To demonstrate this, we now **extend Tables 1,3,4 and 5** with FP8 accuracy measurements and add FP8 end-to-end latency evaluations to **Appendix G**, where we demonstrate up to 1.3x inference speedup of FP4 over the latter.
>
>
> > The methodological novelty is moderate, as MR-GPTQ mainly adapts the existing GPTQ framework
>
> We believe that the main novelty of our work is in the formal analysis of the quantization formats, and the resulting practical micro-rotation proposal, which leads to state-of-the-art results for both NVFP and MXFP.
>
> In that context, it is useful to see the design of MR-GPTQ as the logical continuation of the insights obtained from that analysis:
>  * We started by observing that SOTA transform-based W&A quantization methods (e.g., QuaRot) hurt accuracy when coupled with the NVFP4 format, but that the degradation decreases as we decrease the transform block size, with block size matching quantization group size being roughly optimal.
>  * This led to our theoretical analysis, showing that this behaviour is coupled with the expected quantization error (MSE) for the corresponding weight and activation distributions.
>  * Finally, this inspired our new variant of GPTQ (whose main guiding principle is to minimize MSE).
>  * The process for the MXFP process was very similar, with the key difference being the fact that the scale format leads it to benefit from larger transform sizes (e.g. 128 vs 16).
>
> We will update the discussion in Section 4.1 to further clarify this sequence of steps.
>
> > The current speed evaluation focuses mainly on end-to-end throughput. Is there any latency measurements for the prefill and decode stages, respectively?
>
> **We now add Appendix G** “QuTLASS Speedups in Various Inference Regimes” where we show **1.2x and 1.3x end-to-end speedup** of FP4 kernels over FP8 and BF16 baselines for small- and large- batch inference respectively.
>
> Please note that the kernels we provided in the submission were not designed specifically for small-batch inference, and, as such, had lower speedups in that regime. Additionally, it would not make sense to utilize activation quantization for small batch sizes, as this is a memory-bound inference regime, for which quantizing activations does not provide significant speedup. Instead, it would make more sense to use weight-only quantization. Please recall that we provided accuracy results for MR-GPTQ in the weight-only case in Appendix A of our submission, showing that our approach is equally effective as well. We have added latency measurements and discussion for the low-batch regime / weight-only quantization in **Appendix G** as well. In short, we observe higher speedups at small batch sizes for the weight-only kernel, which naturally decrease in the large batch size case.

---

### Official Review · Reviewer_KK3i · 2025-10-27

**Soundness:** 3
**Presentation:** 3
**Contribution:** 2
**Rating:** 8
**Confidence:** 4

**Summary:**

This paper systematically explores the accuracy–efficiency trade-offs of 4-bit floating-point formats (MXFP4, NVFP4) and introduces QuTLASS, a GPU kernel template that integrates the Hadamard transform at effectively zero additional overhead. The study also examines how INT-family quantization methods interact with FP4 to assess practical benefits for real-world inference.

**Strengths:**

- The paper is well structured and easy to follow.
- Broad empirical exploration of combining INT quantization methods with FP4 (MXFP4/NVFP4).
- Introduction of QuTLASS, which integrates the Hadamard transform with minimal overhead.

**Weaknesses:**

- The text states FP8 is “near-lossless,” yet Table 1 lacks an FP8 baseline (e.g., E4M3/E5M2). Including it under identical conditions would anchor FP4’s position more convincingly.
- Although MR-GPTQ runs offline and does not affect inference latency directly, reproducibility details are missing (GPU type/count, hours). Please add a minimal, reproducible configuration.
- (Minor) L238: typo “teh” → “the.”

**Questions:**

- With FP8 (E4M3/E5M2) included as a baseline under identical tuning, how large and consistent is the gap to FP4 across tasks?
- What is the minimal reproducible setup for MR-GPTQ (GPU count, wall-time, memory)? How does cost scale for small vs. large models?

---

> ### Author Response · Authors · 2025-11-17
>
> > The text states FP8 is “near-lossless,” yet Table 1 lacks an FP8 baseline (e.g., E4M3/E5M2). Including it under identical conditions would anchor FP4’s position more convincingly.
>
> Thank you for the suggestion. We conducted an evaluation of FP8 quantized models following the same protocol adopted for FP4 formats and provide the results in the table below:
>
> | Model | Format | Quantization | MMLU-CoT | GSM8k | Hellaswag | Winogrande | Average | Recovery % |
> |---|---|---|---:|---:|---:|---:|---:|---:|
> | Llama 3.2 1B Instruct | FP16 | N/A | 0.4427 | 0.4632 | 0.5984 | 0.6172 | 0.5304 | - |
> |                                    | FP8 | RTN | 0.4610 | 0.4470 | 0.5950 | 0.5950 | 0.5250 | 98.99 |
> |                                    | FP8 | GPTQ | 0.4580 | 0.4500 | 0.5910 | 0.6060 | 0.5263 | 99.22 |
> |                                    | NVFP4 | MR-GPTQ | 0.3899 | 0.3298 | 0.5666 | 0.5817 | 0.4670 | 87.35 |
> | Llama 3.2 3B Instruct | FP16 | N/A | 0.6471 | 0.7786 | 0.7339 | 0.7017 | 0.7153 | - |
> |                                    | FP8 | RTN | 0.6340 | 0.7770 | 0.7300 | 0.6970 | 0.7095 | 99.19 |
> |                                    | FP8 | GPTQ | 0.6410 | 0.7780 | 0.7300 | 0.6990 | 0.7120 | 99.54 |
> |                                    | NVFP4 | MR-GPTQ | 0.6026 | 0.6876 | 0.7105 | 0.6780 | 0.6697 | 93.68 |
> | Llama 3.1 8B Instruct | FP16 | N/A | 0.7276 | 0.8506 | 0.8001 | 0.7790 | 0.7893 | - |
> |                                    | FP8 | RTN | 0.7240 | 0.8470 | 0.7980 | 0.7770 | 0.7865 | 99.64 |
> |                                    | FP8 | GPTQ | 0.7180 | 0.8450 | 0.7990 | 0.7810 | 0.7858 | 99.55 |
> |                                    | NVFP4 | MR-GPTQ | 0.6912 | 0.8080 | 0.7817 | 0.7524 | 0.7584 | 96.08
>
> We have now updated **Tables 1,3,4 and 5** to include these baselines.
>
> We observe that both 8‑bit formats achieve near-lossless accuracy (about 99.0–99.5% recovery), even on smaller models, which are typically harder to compress. However, 8‑bit quantization limits memory reduction and throughput gains to roughly 2× versus the FP16/BF16 baseline. In contrast, MXFP4/NVFP4 deliver up to 4× savings in memory and inference time. To validate this, **we now add Appendix G** “QuTLASS Speedups in Various Inference Regimes” where we show **1.2x and 1.3x end-to-end speedup** of FP4 kernels over an FP8 baseline for small- and large- batch inference, respectively, in vLLM.
>
>
> > What is the minimal reproducible setup for MR-GPTQ
>
> We conduct most of our experiments using the provided pseudo-quantization Triton implementation on H100 nodes, although the portability of Triton theoretically allows for the use of older GPU architectures too.
>
> For RTN quantization, the process takes just a few seconds and requires the memory occupied by the quantized model in the end (i.e., just 4 GB for 8B models).
>
> For GPTQ quantization and similar methods, quantization of an 8B takes roughly 30 minutes on a single H100 GPU, with larger models taking proportionally more time. The provided GPTQ implementation allows for quantization of models with up to 70B parameters on a single H100 GPU (i.e., under 80 GB of DRAM) due to layer offloading.
>
> For QAT, 8B training took roughly 30 hours on 8xL40S GPUs and as little as 2 hours for 1B models.
>
> In total, we expect everyone who is able to run the models to also be able to produce them themselves, with maybe the exception of QAT, which requires server-grade hardware for full training. We will include this discussion in the next revision of the manuscript.

---

> > ### Comment · Reviewer_KK3i · 2025-11-23
> >
> > I have read the rebuttal and thanks for the additional data and explanation. I maintain my original score.

---

### Official Review · Reviewer_34aN · 2025-11-01

**Soundness:** 3
**Presentation:** 2
**Contribution:** 3
**Rating:** 4
**Confidence:** 4

**Summary:**

This paper proposes MR-GPTQ (Micro-Rotated GPTQ), a post-training quantization scheme tailored for FP4 formats, aiming to overcome the limited accuracy recovery observed when applying existing PTQ methods to FP4-quantized LLMs.

Although recent support for MXFP4 and NVFP4 on Blackwell-class GPUs enables efficient 4-bit inference, the authors show that conventional INT4-oriented techniques do not transfer well to FP4 due to format-specific constraints—most notably, NVFP4’s small group size and MXFP4’s power-of-two scale quantization (E8M0). Moreover, contrary to observations in prior low-bit quantization work, the authors find that Hadamard-based rotations can degrade MSE in NVFP4, exposing a mismatch between FP4 properties and existing rotation-based normalization strategies.

To address these issues, the authors present MR-GPTQ, which adapts GPTQ’s second-order update mechanism to FP4 via block-wise Hadamard rotation, format-aware scale optimization, static activation reordering, and fused GPU kernels (QuTLASS) optimized for Blackwell.

Experimental results on Llama-3 and Qwen models show that NVFP4 combined with MR-GPTQ recovers approximately 98–99% of FP16 accuracy, while MXFP4—despite its inherently larger quantization error—benefits substantially and approaches NVFP4-level performance. In addition, end-to-end inference achieves 2.2×–4× speedups on B200 and RTX5090 GPUs, demonstrating that FP4 can deliver strong accuracy–performance trade-offs when paired with format-specialized PTQ.

**Strengths:**

* Provides a systematic and quantitative analysis of the MXFP4/NVFP4 formats, clearly identifying the root causes of accuracy degradation in FP4 PTQ.
* Demonstrates, both empirically and analytically, that rotation-based PTQ techniques are not consistently effective under FP4, motivating the need for format-aware methods.
* Proposes MR-GPTQ, an FP4-specialized variant of GPTQ, which delivers meaningful accuracy improvements by explicitly accounting for FP4 constraints.
* Introduces Blackwell-optimized QuTLASS kernels that minimize the runtime overhead of micro-rotations, yielding practical performance gains (2.2×–4× end-to-end).
* Presents a well-structured pipeline—from analysis to algorithm design, kernel implementation, and evaluation—resulting in a cohesive and well-justified contribution.

**Weaknesses:**

### Major Weaknesses
* Insufficient motivation for choosing FP4 over INT4 (potential logical gap):
    * The paper adopts FP4 (MXFP4/NVFP4) as the primary target format, yet it does not clearly articulate why FP4 should be preferred over INT4 in PTQ scenarios.
    * A more principled comparison—including distributional properties, quantization behavior, and rotation sensitivity—would help substantiate the motivation for focusing on FP4.
    * If strong INT4 + rotation baselines (e.g., QuaRot/QuIP) can achieve performance comparable to or better than FP4 + MR-GPTQ, the necessity and relative advantage of the proposed approach may become less evident.
* Limited justification and characterization of MR-GPTQ design choices:
    * The paper introduces several components—format-aware scaling, static activation reordering, and block-wise rotation—but does not clearly explain why each is particularly necessary or effective for FP4.
    * A more principled analysis that isolates the contribution of each component (e.g., through ablations or sensitivity) would clarify whether these choices are inherently beneficial, or simply empirical heuristics.
    * It is also unclear under which conditions these components fail, making it difficult to assess the robustness and general applicability of the proposed design.

### Minor Weaknesses
* The source of FP4 rotation degradation is not clearly isolated; the paper does not distinguish format-intrinsic effects from configuration factors (e.g., block size, scaling, calibration), making the necessity of the proposed approach less firmly supported.
* The observed superiority of NVFP4 over MXFP4 is largely expected from format specifications (E4M3 vs. E8M0), so the empirical confirmation offers limited new insight.
* The method primarily applies heuristic extensions to GPTQ without demonstrating a substantial methodological advance, which may make the contribution appear incremental rather than novel.

**Questions:**

If the authors could clarify or further substantiate the major weaknesses mentioned above, I would be happy to reconsider and potentially raise my evaluation. I would particularly appreciate further clarification on these major concerns.

---

> ### Author Response · Authors · 2025-11-17
>
> Thank you for your time and dedication to review this submission. We address your concerns and questions one-by-one.
>
> > Insufficient motivation for choosing FP4 over INT4
>
> The choice to analyze FP4 in our paper is motivated by the fact that FP4 is the only 4-bit format supported on the latest-generation NVIDIA Blackwell GPUs. INT4 operations support has been discontinued since the NVIDIA Ampere family of GPUs. In fact, no NVIDIA GPUs feature support for both FP4 and INT4 operations, making the “choice” of FP4 over INT4 a matter of necessity, rather than preference.
>
> To bridge this analysis gap and inform future hardware designs, we now present a new analysis of _hypothetical_ microscaling INT4 formats, which we analyze and validate in the same context of Sections 3 and 5 of the paper.
> Since INT4 is not a part of the “OCP Microscaling Formats (MX) Specification”, we define it ourselves below:
>  * We define INT4 base data type as a uniform symmetric grid of 15 (to match FP4)  elements: `[-7, -6, -5, -4, -3, -2, -1, 0, 1, 2, 3, 4, 5, 6, 7]` .
>  * We define NVINT4 as a microscaling format with E4M3 shared scales for groups of 16 elements (same as NVFP4) over the INT4 base element data type.
>  * We define MXINT4 as a microscaling format with E8M0 shared scales for groups of 32 elements (same as MXFP4) over the INT4 base element data type.
>
> Applying the quadratic error analysis from Section 3 to these formats reveals that NVINT4 performs close to NVFP4 without the normalizing transforms. Yet, the proposed group Hadamard Transform has positive effect on it (as opposed to negative for NVFP4), making micro-rotated NVINT4 the most accurate of all the analyzed formats. MXINT4, however, performs poorly, and the normalizing transforms yield limited improvement.
>
> Conducting evaluations from Section 5 for NVINT4 and MXINT4, we validate that the Hadamard Transform indeed has a positive effect on these formats and that NVINT4 with the Hadamard Transform is the best-performing tested microscaling format, as predicted by the analysis above.
>
> These results expand our analysis to INT4-based formats and highlight their potential usefulness. We now include them in **Appendix D** “INT4-Based Microscaling Formats”, **Figure 8** and in **Tables 1,3,4 and 5**. We will refine the presentation, as well as include a broader discussion of these potential formats, in the next revision of the manuscript.
>
> > Limited justification and characterization of MR-GPTQ design choices
>
> From ablations perspective, please note that **Appendix H** “Standard Deviation” includes the detailed breakdown of MR-GPTQ performance that tests its components (e.g., scaling optimization, pass order and transform sizes) in isolation and provides statistical significance estimations for their effects.
>
> From design perspective, the following line of thought led to its final form:
>  * We started by observing that SOTA transform-based W&A quantization methods (e.g., QuaRot) hurt accuracy when coupled with the NVFP4 format, but that the degradation decreases as we decrease the transform block size, with block size matching quantization group size being roughly optimal.
>  * This led to our theoretical analysis, showing that this behaviour is coupled with the expected quantization error (MSE) for the corresponding weight and activation distributions.
> Finally, this inspired our new variant of GPTQ (whose main guiding principle is to minimize MSE).
>  * The process for the MXFP process was very similar, with the key difference being the fact that the scale format leads it to benefit from larger transform sizes (e.g. 128 vs 16).
>
> We will update the discussion in Section 4.1 to further clarify this sequence of steps.
>
> > Minor Weaknesses
>
> > The observed superiority of NVFP4 over MXFP4 is largely expected…
>
> Firstly, it was not obvious to us, a priori, that NVFP4 has enough dynamic range to capture all the shared scales in LLMs. In fact, it does not, and we had to introduce additional per-tensor scales to align shared scales distributions and verify that they fall within the E4M3 range, as shown in Figure 4. MXFP4 does not have this downside and does not require per-tensor re-scaling. From this perspective, it is more of a plug-and-play solution than NVFP4.
>
> Secondly, NVFP4 has both higher average bitwidth (4.50 vs 4.25) and slower throughput (see Figure 5) than MXFP4, making the choice between the two a non-obvious trade-off.
> The combination of micro-rotations and GPTQ that we propose reduces the performance gap between MXFP4 and NVFP4, on average, by more than 60% (Figure 4), affecting the trade-off balance. QAT results show even larger performance gap reduction (80% on average).
>
> Thus, we gently push back on the statement that NVFP4’s superiority is to be expected: in our view, the two formats present an accuracy-performance trade-off.

---

> > ### Comment · Reviewer_34aN · 2025-11-27
> >
> > Thank you for the authors’ thoughtful and detailed responses.
> >
> > After carefully reviewing the rebuttal, the clarifications provided to other reviewers, and the revised manuscript, I find that my main concerns, as well as most of my remaining questions, have been adequately addressed.
> >
> > Accordingly, I have decided to increase my overall score. I would like to once again express my appreciation for the authors’ clear and constructive responses.

---

### Official Review · Reviewer_2iLw · 2025-11-03

**Soundness:** 3
**Presentation:** 3
**Contribution:** 3
**Rating:** 6
**Confidence:** 3

**Summary:**

This paper analyzes the quantization error in the NVFP4 and MXFP4 formats and shows that rotation-based quantization could help MXFP4 but hurts the accuracy of NVFP4 when using round-to-nearest quantization. A new quantization method, MR-GPTQ, is proposed to use block-wise Hadamard micro-rotations, static activation reordering, and format-specific optimizations to improve the accuracy of both NVFP4 and MXFP4. The evaluation results show that MR-GPTQ could achieve better accuracy recovery than prior art. The performance results show the overhead introduced by MR-GPTQ is very small, and 2.2x and 4x end-to-end speedup is demonstrated on server-grade (B200) and consumer-grade GPUs (RTX5090).

**Strengths:**

- A good theoretical analysis and empirical validation of quantization error on the latest quantization schemas (NVFP4 and MXFP4).
- A good kernel implementation that minimizes the overhead of the proposed quantization method and demonstrates strong end-to-end speedup on cutting-edge GPUs.

**Weaknesses:**

- It would be helpful if the proposed method could be compared against existing quantization methods, FP8 and INT8, which is already used in many production situations.

**Questions:**

- How does the proposed method compare to FP8 and INT8 in quality and end-to-end performance?
- There is a gap between single-layer speedup and end-to-end speedup. Could you provide some analysis of this? Also, in Figure 5 (right), why is the speedup of FP4 low in a small batch size? I think attention is not a bottleneck here since the sequence length is not very long.

---

> ### Author Response · Authors · 2025-11-17
> **New FP8/INT8 evaluations. New kernel benchmarks.**
>
> Thank you for assessing our work. We are happy to provide answers to the questions you asked.
>
> > How does the proposed method compare to FP8 and INT8…
>
> To address your question, we evaluated INT8 and FP8 quantized models following the same calibration protocol as FP4 models:
>
> | Model | Format | Quantization | MMLU-CoT | GSM8k | Hellaswag | Winogrande | Average | Recovery % |
> |---|---|---|---:|---:|---:|---:|---:|---:|
> | Llama 3.2 1B Instruct | FP16 | N/A | 0.4427 | 0.4632 | 0.5984 | 0.6172 | 0.5304 | - |
> |                                    | INT8 | RTN | 0.4590 | 0.4420 | 0.5980 | 0.6130 | 0.5280 | 99.55 |
> |                                    | INT8 | GPTQ | 0.4540 | 0.4490 | 0.5960 | 0.6010 | 0.5250 | 98.99 |
> |                                    | FP8 | RTN | 0.4610 | 0.4470 | 0.5950 | 0.5950 | 0.5250 | 98.99 |
> |                                    | FP8 | GPTQ | 0.4580 | 0.4500 | 0.5910 | 0.6060 | 0.5263 | 99.22 |
> |                                    | NVFP4 | MR-GPTQ | 0.3899 | 0.3298 | 0.5666 | 0.5817 | 0.4670 | 87.35 |
> | Llama 3.2 3B Instruct | FP16 | N/A | 0.6471 | 0.7786 | 0.7339 | 0.7017 | 0.7153 | - |
> |                                    | INT8 | RTN | 0.6400 | 0.7770 | 0.7330 | 0.6960 | 0.7115 | 99.47 |
> |                                    | INT8 | GPTQ | 0.6400 | 0.7790 | 0.7340 | 0.6990 | 0.7130 | 99.67 |
> |                                    | FP8 | RTN | 0.6340 | 0.7770 | 0.7300 | 0.6970 | 0.7095 | 99.19 |
> |                                    | FP8 | GPTQ | 0.6410 | 0.7780 | 0.7300 | 0.6990 | 0.7120 | 99.54 |
> |                                    | NVFP4 | MR-GPTQ | 0.6026 | 0.6876 | 0.7105 | 0.6780 | 0.6697 | 93.68 |
> | Llama 3.1 8B Instruct | FP16 | N/A | 0.7276 | 0.8506 | 0.8001 | 0.7790 | 0.7893 | - |
> |                                    | INT8 | RTN | 0.7250 | 0.8480 | 0.8020 | 0.7740 | 0.7873 | 99.74 |
> |                                    | INT8 | GPTQ | 0.7240 | 0.8440 | 0.8000 | 0.7730 | 0.7853 | 99.48 |
> |                                    | FP8 | RTN | 0.7240 | 0.8470 | 0.7980 | 0.7770 | 0.7865 | 99.64 |
> |                                    | FP8 | GPTQ | 0.7180 | 0.8450 | 0.7990 | 0.7810 | 0.7858 | 99.55 |
> |                                    | NVFP4 | MR-GPTQ | 0.6912 | 0.8080 | 0.7817 | 0.7524 | 0.7584 | 96.08
>
> We have now updated **Tables 1,3,4 and 5** to include these baselines.
>
> We observe that both 8‑bit formats achieve near-lossless accuracy (about 99.0–99.5% recovery), even on smaller models, which are typically harder to compress. However, 8‑bit quantization limits memory reduction and throughput gains to roughly 2× versus the FP16/BF16 baseline. In contrast, MXFP4/NVFP4 deliver up to 4× savings in memory and inference time. To validate this, **we now add Appendix G** “QuTLASS Speedups in Various Inference Regimes” where we show **1.2x and 1.3x end-to-end speedup** of FP4 kernels over an FP8 baseline for small- and large- batch inference, respectively, in vLLM.
>
> > There is a gap between single-layer speedup and end-to-end speedup. Could you provide some analysis of this? I think attention is not a bottleneck here since the sequence length is not very long.
>
> The performance gap can be explained by the fact that non-linear operations and as well as attention have non-zero cost, as well as the input processing times and scheduling engine overhead. These are automatically accounted for in our end-to-end measurements, and limit the speedup impact of lower-precision matrix multiplies.
>
> > In Figure 5 (right), why is the speedup of FP4 low in a small batch size?
>
> The kernels we provided in the submission were not designed specifically for small-batch inference, and, as such, had lower speedups in that regime. Additionally, it would not make sense to utilize activation quantization for small batch sizes, as this is a memory-bound inference regime, for which quantizing activations does not provide significant speedup. Instead, it would make more sense to use weight-only quantization. Please recall that we provided accuracy results for MR-GPTQ in the weight-only case in Appendix A of our submission, showing that our approach is equally effective as well.
> To fully address your question, we have added latency measurements and discussion for the low-batch regime / weight-only quantization in **Appendix G** “QuTLASS Speedups in Various Inference Regimes”. In short, we observe higher speedups at small batch sizes for the weight-only kernel, which naturally decrease in the large batch size case.

---

### Comment · Area_Chair_JFb5 · 2025-11-20
**Action Needed: Review Rebuttal and Update Evaluation**

Dear Reviewers,

Thank you, as always, for your valuable contributions and efforts. The authors have now submitted their rebuttal. Please take a moment to review it and provide any necessary follow-up actions, such as additional questions, clarification requests, or updates to your review.

Since the initial ratings ranged from 8 to 4, I kindly ask you to pay close attention to the perspectives of the other reviewers when preparing your final response.

Thank you again for your support.

---

### Author Response · Authors · 2025-12-02
**Summary of the rebuttal**

Dear Area Chair

We sincerely thank the reviewers and area chair for the constructive feedback on our submission. During the rebuttal, one of the reviewers (**34aN**) increased their score from **4 to 6** after their concerns were addressed. Below, we summarize the main concerns raised and how they were resolved during the discussion.

---

**Question**: Comparison with FP8

**Answer**: Reviewers  **2iLw** and  **KK3i** requested a comparison with FP8 in terms of accuracy and inference speed. We conducted an evaluation of FP8-quantized models under the same experimental setup as for FP4 formats. The results show that FP8 is essentially lossless in accuracy, whereas FP4 incurs a certain performance drop. However, FP4 achieves faster inference across a wide range of batch sizes, making it preferable in latency constrained setting.

---

**Question**: Comparison with INT

**Answer**: Reviewer  **34aN** asked whether a floating-point representation is preferable to integer quantization. Our primary motivation for focusing on FP4 formats is the hardware support in the latest generation of NVIDIA GPUs for microscaling MXFP/NVFP formats, as well as the lack of efficient low-precision integer operations in the Blackwell generation.

Following the reviewer’s suggestion, we additionally produced INT4-quantized models with microscaling and observed that integer formats perform at least as well as, and in some cases slightly better than, the corresponding floating-point formats.

---

**Question**: Justification of MR-GPTQ design choices and the methodological novelty

**Answer**: Reviewers **34aN** and **WXdo** asked for a clearer justification of the specific MR-GPTQ design choices and the methodological novelty. In our study, we start from the observation that established transform-based techniques for quantization error reduction do not perform well when combined with microscaling formats. We show empirically that using large transform sizes can harm accuracy, and that setting the rotation size equal to the group size is optimal.
We then provide a theoretical analysis of the expected quantization error. This analysis, together with the empirical findings, motivates a new variant of GPTQ whose main guiding principle is to minimize the mean squared error at the layer output.

---

**Question**: Latency measurements for prefill and decode setup.

**Answer**: Reviewer **WXdo** requested latency measurements for both the prefill and decode stages. Following this suggestion, we ran inference benchmarks for both small and large batch sizes. We observed end-to-end speedups of **1.2×** and **1.3×**, respectively, for FP4 kernels compared to the FP8 baseline.

---

### Meta-Review · Area_Chair_4MMs · 2025-12-24

**Summary:**

This paper considers W4A4 inference with micro scaling, with a set of comprehensive accuracy experiments on state-of-the-art open source models, kernel implementations, new findings, and new quantization algorithms. Reviewers have a consensus towards acceptance.

**Reviewer Concerns:**

1. end-to-end speedup
2. motivation of micro scaled FP4 over traditional INT4
3. the methodology is largely a combination of existing techniques
4. the final method is still lossy

Overall, most reviewer concerns have been adequately addressed.

**Reviewer Scores:**

Initial rating was (8,4,6,8). As most concerns are addressed, I expect the rating-4 reviewer would increase to 6.

---

### Decision · Program_Chairs · 2026-01-26

Accept (Poster)